# Quantifying bioalbedo: A new physically-based model and discussion of empirical methods for characterizing biological influence on ice and snow albedo

Joseph M Cook[1,2], Andrew J Hodson[1,3], Alex S Gardner[4], Mark Flanner[5], Andrew J Tedstone[6], Christopher Williamson[6], Tristram DL Irvine-Fynn[7], Johan Nilsson[4], Robert Bryant[1], Martyn Tranter[6]

[1] Department of Geography, University of Sheffield, Winter Street, Sheffield, UK

[2] College of Life and Natural Sciences, University of Derby, Kedleston Road, Derby, UK

[3] University Centre in Svalbard (UNIS), Svalbard, Norway

[4] Jet Propulsion Laboratory, California Institute of Technology, Pasadena, USA

[5] Department of Climate and Space Sciences and Engineering, University of Michigan, Ann Arbor MI 48105, USA

[6] Centre for Glaciology, University of Bristol, Bristol, UK

[7] Centre for Glaciology, Aberystwyth University, Aberystwyth, Wales

*Correspondence to*: Joseph M Cook (joe.cook@sheffield.ac.uk)

**Abstract.**

The darkening effects of biological impurities on ice and snow have been recognized as a control on the surface energy balance of terrestrial snow, sea ice, glaciers and ice sheets. With a heightened interest in understanding the impacts of a changing climate on snow and ice processes, quantifying the impact of biological impurities on ice and snow albedo ('bioalbedo') and its evolution through time is a rapidly growing field of research. However, rigorous quantification of bioalbedo has remained elusive because of difficulties isolating the biological contribution to ice albedo from that of inorganic impurities and the variable optical properties of the ice itself. For this reason, isolation of the biological signature in reflectance data obtained from aerial/orbital platforms has not been achieved, even when ground-based biological measurements have been available. This paper provides the cell specific optical properties that are required to model the spectral signatures and broadband darkening of ice. Applying radiative transfer theory, these properties provide the physical basis needed to link biological and glaciological ground measurements with remotely sensed reflectance data. Using these new capabilities we confirm that biological impurities can influence ice albedo then identify ten challenges to the measurement of bioalbedo in the field with the aim of improving future experimental designs to better quantify bioalbedo feedbacks. These challenges are: 1) Ambiguity in terminology, 2) Characterizing snow or ice optical properties, 3) Characterizing solar irradiance, 4) Determining optical properties of cells, 5) Measuring biomass, 6) Characterizing vertical distribution of cells, 7) Characterizing abiotic impurities, 8) Surface anisotropy, 9) Measuring indirect albedo feedbacks, and 10) Measurement and instrument configurations. This paper aims to provide a broad audience of glaciologists and biologists with an overview of radiative transfer and albedo that might be guide future experimental design.

## 1 Background:

The presence of biological impurities in the cryosphere has been known for more than a century with scientific interest dating back to 1676 when Van Leeuwenhoek described microbial life in snow samples. In the late 19th Century, Nordenskiöld (1875) coined the term 'cryoconite' to describe the distinct biological aggregates that were visible on the ice surface of the Greenland Ice Sheet and noted the potential for biological activity to amplify melt, stating that it was the 'greatest enemy to the mass of ice' and a potential accelerator of deglaciation. Since the late twentieth century, understanding snow and ice melt processes has gained new significance because loss of terrestrial ice is increasingly contributing to sea level rise and driving climate-

amplifying positive feedbacks, many of which are linked to snow and ice reflectivity (Budyko 1969; Flanner et al. 2011; IPCC, 2013; Hanako and Nazarenko, 2004). The direct impact of biological material on snow and ice albedo has recently become known as 'bioalbedo', following an abbreviation of the term "biological albedo" presented by Kohshima et al., (1993).

There are two distinct areas of research into the biological darkening of the cryosphere: the first examines snow algal blooms (green and red patches) in seasonal snowpacks, and the other has focused on ice algal blooms on ice ablation zones. Ground reflectance and aerial remote sensing have been used to quantify algae in snow in the North American Sierra Nevada (Painter et al., 2001). The same research group has also developed techniques for deriving optically equivalent grain size and specific surface area from spectral reflectance data that has proven useful for satellite observations of aging snow (Painter et al., 2007;

2013). Satellite retrieval of snow algal discoloration in the Harding Ice Field has also been achieved using similar techniques by Takeuchi et al. (2006). Both groups used different methods for quantifying microbial abundance in snow from remotely sensed reflectance data: Painter et al. (2001) used the integral of a chlorophyll-specific absorption feature that relies upon the availability of data with sufficiently high spectral resolution, whereas Takeuchi et al. (2006) used a broader carotenoid absorption feature as a diagnostic biomarker. Other groups have examined snow algae from a biological perspective. Remias

et al. (2005; 2010) showed that snow algae produce pigments in response to environmental stresses that changes their colour, and thereby influences snow albedo. Hodson et al. (*in press*) also documented how both surface and sub-surface algal populations influenced the optical properties of maritime Antarctic snow.

    Glacier albedo can also be modified by ice algae, which are taxonomically distinct from the algae that inhabit snow (Uetake

et al., 2010; Yallop et al., 2012). Ice algae are dominated by several species of green algae and cyanobacteria. The relative abundance of different species of ice algae and cyanobacteria vary spatially: the green alga *Mesotaenium berggrenii* has been shown to dominate near the termini of Qaanaaq Glacier (north-west Greenland: Uetake et al. 2010) and Tyndall Glacier (Patagonia: Takeuchi and Kohshima, 2004) and is also present in western Greenland (Yallop et al. 2012). *Ancylonema nordenskioldii* was found to dominate the upper ablation areas of Gulkana Glacier (Alaska: Takeuchi et al. 2009) and Qaanaaq

Glacier (Uetake et al. 2010) and spread extensively over the western Greenland Ice Sheet along with *Cylindrocystis spp.* (Yallop et al. 2012). An abundance of both *Ancylonema* and *Cylindrocystis* on the Greenland Ice Sheet was reported in the nineteenth century (Berggren, 1871; Nordenskiöld, 1875) and mid twentieth century (Gerdel and Drouet, 1960).

    The albedo reducing effect of ice algae arises from their dark pigmentation. This was discussed by Yallop et al. (2012) who

linked algal pigmentation to ice albedo in south-west Greenland, but did not explicitly identify the pigments responsible. Ice algae produce a range of pigments to facilitate both light harvesting across the visible spectrum and energy dissipation under conditions of excessive irradiance (Remias et al., 2009), via ,for example, the interconversion of xanthophyll pigments to quench excess excitation energy as heat (Niyogi et al., 1997; Goss and Jakob, 2010). In addition, ice algae inhabiting surface habitats produce specialist UV-absorbing pigments (purpurogallins) that are distinct from the typical UV-absorbing pigments

(mycosporine-like amino acids) identified in other algal lineages (Remias et al., 2012). Yallop et al., (2012) found these pigments to obscure the primary light harvesting and protective pigments and to lower the albedo of the ice sheet surface.

    The wide spatial coverage of algae on the ablating areas of the Greenland Ice Sheet (which, over the past 15 years have comprised between 5 – 16% of the entire ice sheet area: Shimada et al., 2016) combined with their strong absorption of visible

light (which accounts for approximately half of the solar energy incident upon the ice surface: Liou, 2002) makes them important agents of albedo reduction and therefore melt accelerators. However, empirical studies that link ice algal biomass to albedo are scarce. To date, published measurements have been limited to broadband albedos measured using opposed pyranometers without quantification of the physical properties of the ice surface, nor its inorganic impurity content (Yallop et

al., 2012; Lutz et al., 2014). Therefore, the biological contribution to ice albedo ('bioalbedo') and its spatiotemporal variability remains unquantified. In this paper, we will show that it cannot be isolated from non-biological albedo reducing processes without refinements to current empirical techniques and without the use of a spectrally resolved radiative transfer model.

Weak, often semi-quantitative correlations have been reported between cell numbers and albedo (e.g. Lutz et al., 2016) which is unsurprising since the majority of the variation in albedo is due to physical characteristics of the snow or ice including internal factors such as ice crystal size and water content, as well as external factors such as solar angle, atmospheric effects, shading and multiple reflections between surface roughness elements (Gardner and Sharp, 2010). Takeuchi (2002), Takeuchi and Kohshima (2004) and Takeuchi et al., (2015) showed the albedo reduction resulting from biotic and abiotic impurities to
vary between different glaciers, signifying that the mass and optical properties of abiotic impurities such as dusts are also crucial determinants of surface albedo. This may be particularly relevant on the western Greenland Ice Sheet, where high concentrations of dust may be outcropping from melting Holocene ice (Bøggild et al., 2010; Wientjes et al., 2010; 2011). The vertical position of light absorbing impurities in the shallow-subsurface is also important because in the visible wavelengths absorption by ice crystals is low (Warren, 1982), meaning light can penetrate the snowpack. These complexities make the
interpretation of previous results difficult. For these reasons, simple empirical models linking albedo to microbial abundance on glacier ice will have small coefficients of determination and leave a large portion of the variance in albedo unexplained. To understand bioalbedo, a physical modelling approach is required to link ground measurements to spectral data and to integrate effects of both biological and non-biological impurities being present in and on the ice. Physical modelling establishes a functional relationship between snow and ice properties and surface radiance, enabling both predictions of albedo change given
changes in optical properties and surface parameter retrieval from spectral reflectance, and therefore allows us to quantify the bioalbedo of a complex mixture of ice, biotic and abiotic impurities.

Aoki et al. (2013) and Cook et al. (2017) have previously used radiative transfer models to suggest algal cells can directly influence the melt rate of a snowpack due to the bioalbedo effect, although these models simply updated the absorption terms
in existing inorganic impurity models. Here we improve upon their models by creating a library of optical properties for algal cells using Mie theory and coupling this to an adapted form of the SNICAR radiative transfer scheme (Flanner et al., 2007). This new physical bioalbedo model is then used to support a discussion of appropriate experimental designs for future empirical bioalbedo studies. In the first section, a brief review of radiative transfer modelling of snow and ice will precede the presentation of the model, specifically aimed a broad audience likely comprising glaciologists and biologists rather than
spectroscopists or remote sensors. Then, the discussion of empirical bioalbedo studies will proceed by detailed examination of ten distinct challenges whose effects are quantified using the model where possible.

**2 Radiative Transfer modelling**

Several radiative transfer models (RTMs) exist for snow, the most well-known being the two-stream radiative flux models SNICAR (Flanner et al., 2007) and TARTES (Libois et al., 2013), and the plane-parallel multi-stream DIScrete Ordinates Radiative Transfer scheme (DISORT, Stamnes et al., 1988) used by the MODTRAN (Berk et al., 2014) and SBDART (Ricchiazzi et al., 1998) atmospheric transmission models. The two stream approach is computationally efficient and able to accurately predict spectral reflectance hemispherically; however, a multi-stream approach is required to predict directional
reflectance. These models are well-validated for clean snow (Grenfell et al., 1994) and have been applied to snow contaminated with black carbon and dust (Painter et al., 2007, Flanner et al., 2007; Gardner and Sharp, 2010, Brandt et al., 2011; Kaspari et al., 2015), although their performance for impurity-laden snow has been questioned (Skiles et al., 2017). Equivalent radiative transfer schemes for glacier ice are scarce, perhaps due to the diverse range of glacier surface types. Smooth ice is best

represented as a bulk medium of ice with air bubbles and cracks of known size distribution and specular reflection from the upper surface (Gardner and Sharp, 2010; Dadic et al., 2013). In contrast, weathered ice is best described as a collection of ice grains in a bulk medium of air (or water in the case of saturated ice) and is therefore more optically similar to very coarse snow. We suggest that, since algal blooms are a feature of weathered, ablating ice in summer, an adapted snow model is more appropriate for most bioalbedo studies. RTMs are often simplified by assuming ice grains (or air bubbles) to be spherical and homogeneous, such that Mie theory can be used to determine absorption and scattering in the snowpack. This requires a priori knowledge of spectral absorption and scattering coefficients of ice, which have been well known since the 1980's (Warren, 1982) and recently up dated (Warren and Brandt, 2008: Picard et al., 2013). The effects of impurities have been studied many times with particular emphasis on black carbon and certain mineral dusts (Deems et al., 2013; Kaspari et al., 2014; Skiles et al., 2017). Aoki et al. (2013) and Cook et al. (2017) have attempted to incorporate biological impurities into radiative transfer schemes. Aoki et al.'s (2013) model was not fully predictive as it required field spectra to back-calculate absorption coefficients while Cook et al.'s (2017) model lacked field validation. Both models characterized biological impurities simplistically by updating the absorption terms in existing inorganic impurity models rather than predicting the full optical properties of cells using Lorenz-Mie calculations. To date no published datasets include both reflectance at high spectral resolution and sufficient biological and physical metadata to properly validate radiative transfer models for algal snow or ice. Here we address this knowledge gap by coupling a bio-optical model that populates a lookup library of biological impurities created using Mie theory with an adapted form of the two-stream radiative transfer model SNICAR (Flanner et al., 2007). We refer to these coupled models as 'BioSNICAR'.

## 3 BioSNICAR

BioSNICAR is composed of two discrete units: a) a model that calculates optical properties of algal cells; b) a two-stream radiative transfer scheme that accesses a library of biological impurities populated using a). The library comprises NetCDF files containing cell-specific information as presented in Table 1. There are two stages to obtaining the optical properties of cells. First, the complex refractive index of the cells must be known. The real part of the refractive index (representing refraction) is assumed to be constant for all cells at a value of 1.5 (Pottier et al., 2005) whereas the imaginary part of the refractive index ($k_\lambda$, representing absorption) varies according to the pigmentation of the cells. This is modelled using a bio-optical scheme developed using Pottier et al.'s (2005) study of algal cells in bioreactors (also used by Cook et al., 2017) from user-defined values of pigment abundance (as % total cellular dry mass) according to:

$$k_\lambda = \frac{\lambda}{4\pi} \rho_{dm} \frac{1-x_w}{x_w} \sum_{i=1}^{N} Ea_i(\lambda) w_i \quad \text{(Eq. 1)}$$

where

$$x_w = 1 - \frac{C_x}{N_T} \frac{1}{V_{32}} \frac{1}{\rho_{dm}} \quad \text{(Eq. 2)}$$

All terms are defined in Table 1. The water fraction ($x_w$) and density of cellular dry mass ($\rho_{dm}$) are assumed constant throughout the simulations presented in this paper ($\rho_{dm} = 1400$ kg m$^{-3}$, $x_w = 0.8$: Dauchet et al., 2015); however, we provide Equation 2 to enable derivation of $x_w$ from empirical data (see Pottier et al., 2005; Kandilian et al., 2016). It should be noted that there are limits of applicability for Equation 1, since $k_\lambda$ will become infinite when $x_w = 0$. Equation 1 is therefore applicable for $0 < x_w < 1$, where realistic values are likely to be close to 0.8 (Dauchet et al., 2015).

Information about the size of the cells is obtained using a second model that creates particle size distributions from the mean cell radius and standard deviation. This information can be obtained from microscopy of field samples or varied for model experimentation (e.g. Takeuchi et al., 2006). From this information the model calculates all the size information required for

the optical properties of the biological impurities to be obtained using Mie theory. Mie theory utilizes the size information and the complex refractive index to predict the spectral mass absorption cross section, single scattering albedo (defined as the ratio of scattering efficiency to total extinction efficiency) and scattering asymmetry parameter. We used this approach to generate a library of single cell biological impurity optical properties for a range of sizes and pigment compositions that can be updated as required. Similar methodologies have previously been employed to infer the imaginary part of the refractive index of sediments (light et al, 1998) and hydrated salt crystals (Carns et al., 2016) in sea ice and for dust in snow (Skiles et al., 2017).

| Parameter | Unit | Symbol | Wavelength dependent? |
|---|---|---|---|
| Cell radius | m | r | N |
| Mass extinction cross section | $m^2 \, kg^{-1}$ | MAC | Y |
| Single scattering albedo | dimensionless | $\omega_0$ | Y |
| Scattering asymmetry parameter | dimensionless | g | Y |
| Effective surface area-weighted cell radius | m | $R_{surf}$ | N |
| Number mean radius | m | $R_{nmean}$ | N |
| Geometric standard deviation of lognormal cell distribution | dimensionless | $SD_{geo}$ | N |
| Particle Density | $kg \, m^{-3}$ | $\rho_{particle}$ | N |
| Imaginary part of refractive index | dimensionless | k | Y |
| Wavelength | μm | $\lambda$ | N |
| In-vivo mass absorption coefficient | $m^2 \, kg^{-1}$ | Ea | Y |
| Pigment | n/a | i | n/a |
| Density of cellular dry mass | $kg \, m^{-3}$ | $\rho_{dm}$ | N |
| Water fraction in cell | dimensionless | $x_w$ | N |
| Dry mass concentration | $g \, L^{-1}$ in culture suspension | $C_x$ | N |
| Number of pigments | number | N | N |
| Number particle density | $m^{-3}$ | $N_T$ | N |
| Mean efficient volume for particle | $m^3$ | $V_{32}$ | N |
| Mass fraction of pigment | % total cell dry mass | $w_i$ | N |
| Surface area of ice sphere | $m^2$ | S | N |
| Mass of ice sphere | kg | M | N |
| Volume of ice sphere | $m^3$ | V | N |
| Effective radius (i.e. radius of ice spheres with same SSA as the snow or ice) | m | $r_{eff}$ | N |

Table 1: Table of model parameters with their symbols and units. Parameters included in the NETCDF files in the impurity library are shaded grey.

The library of biological impurities (Appendix 1) is coupled to an adapted form of the two-stream radiative flux model SNICAR (Flanner et al., 2007). The mass concentration of cells ($g^{algae}/g^{ice}$) is prescribed in each of *n* layers in a vertical ice column, which is assumed infinite and homogeneous in the horizontal dimension (plane-parallel). The model allows mixing of any number of biological impurities, dusts and black carbon species within each layer provided the relevant optical properties have been calculated and added to the impurity library. By default, SNICAR derives incoming spectral irradiance using an atmospheric RTM (Zender et al., 1997) for either a mid-latitude or Greenland site during either clear sky or cloudy conditions in winter. For simulating albedo at a particular location, the incoming spectral irradiance can be provided to SNICAR as a dataset derived from field measurements or modelling using an atmospheric radiative transfer scheme such as

SBDART (Ricchiazzi et al., 1998) or COART (Jin et al., 2006). For this paper, we used the mid-latitude site under clear-sky conditions (constant solar zenith angle of 60°) as our simulations are not for a particular field site. We have also produced a lookup library of Mie optical properties for a range of snow/ice grain radii, from 10 µm to 5 cm, and the model is able to simulate interstitial melt water using two methods: (a) spheres of liquid water interspersed in the ice matrix (we include this for completeness but note that this introduces additional scattering surfaces and therefore does a poor job of simulating interstitial water) or (b) as liquid water coatings around the ice grains (the preferred method).

## 4 Empirical studies

Modelling enables quantification of changes in albedo resulting from variations in individual components of the ice-atmosphere system. BioSNICAR facilitates this for ice and snow contaminated by biological and non-biological impurities. In this section, we use this capability to identify ten specific challenges to empirical studies of bioalbedo and, where applicable, quantify their impact upon ice albedo using BioSNICAR.

### 4.1 Reconciling ambiguous terminology:

There is an abundance of terminology related to surface reflectance properties in the literature, some of which is ambiguous or used erroneously (Schaepmann-Strubb et al., 2006). Here, we define the most important concepts in an effort to standardize the terminology used in bioalbedo studies. The albedo of a surface is the survival probability of a photon incident on the surface from above (Grenfell, 2011). For snow and ice where emission at long wavelengths (> 4 µm) is negligible, albedo can usually be well approximated as the ratio of upwelling irradiance from a surface to the downwelling irradiance incident upon that surface. This can be for a specific wavelength ("spectral albedo") or integrated over visible and IR wavelengths, weighting by the incoming spectral irradiance ("broadband albedo"). Irradiance is the radiative flux incident upon, emitted from or passing through a surface measured in W m$^{-2}$, whereas radiance is a directional measure of the radiative flux per unit area over a particular solid angle, measured in W m$^{-2}$ sr$^{-1}$. Spectral irradiance is irradiance measured per unit wavelength with units Wm$^{-2}$ µm$^{-1}$. The term 'bioalbedo' is frequently used to describe the impact of biological activity on the albedo of snow and ice.

For an isotropically scattering (Lambertian) surface, incoming solar radiation is scattered equally in all directions in the overlying hemisphere such that reflected radiance in any one direction is equal to that in all other directions. However, all natural surfaces exhibit some degree of anisotropy (Schaepman-Strub et al., 2006). Snow and ice preferentially scatter light in the forward direction, although the angular distribution depends upon lighting geometry, ice optical properties, surface roughness and wavelength (Aoki et al., 2000; Dumont et al., 2010; Hudson et al., 2006). This means that albedo measurements must be made using instruments that sense over a full hemisphere, or alternatively directional measurements must be integrated over a hemisphere weighted by the cosine of the nadir angle (Milton et al., 2009), taking into account the surface bidirectional reflectance distribution function (BRDF). The BRDF describes the amount of radiation reflected in each direction in the overlying hemisphere. Technically, the BRDF is also a conceptual quantity because it is defined by infinitesimal illumination and viewing angles. However, the BRDF can be well-approximated using measurements or simulations at fine angular resolution (Schaepman-Strub et al., 2006) although this is difficult under natural illumination because even in clear sky conditions a large fraction of the UV and blue light is diffuse. Snow and ice albedo is also sensitive to the direction of incoming solar irradiance. For example, at oblique angles a photon can be thought of as 'skimming' the shallow subsurface of a snowpack and is therefore more likely to scatter back out above the snow/ice, whereas a photon entering perpendicular to the surface will experience downward forward scattering deeper into the snowpack, reducing the likelihood of the photon scattering back across the ice-atmosphere interface (Gardner and Sharp, 2010). It is also important to consider whether incoming irradiance is

direct or diffuse. Direct irradiance arrives at a surface having travelled in a straight line from its source, without scattering. Diffuse incoming light, having been scattered at least once from atmospheric particles and molecules, arrives at a surface from all directions within the overlying hemisphere. Truly diffuse light is therefore isotropic. These considerations lead to several possible interpretations of reflectance measurements that have been arranged into a taxonomy by Nicodemus et al. (1977: Fig.

1A). In brief, field measurements using pyranometers or spectral radiometers with cosine-collectors can usually be best described as bihemispheric reflectance (i.e. 'broadband albedo' if the waveband is sufficiently broad). Spectral measurements made under natural illumination using a collimated fibre optic or multispectral imaging system are usually best described as hemispherical conical reflectance factor (HCRF, which approximates to hemispheric directional reflectance factor if the viewing lens collimates to a small solid angle, or to bidirectional reflectance distribution function if both the incoming and

reflected radiance collimate to a small solid angle). Under natural conditions HCRF is similar to BRDF at wavelengths > 0.8 µm, but they are increasingly different at shorter wavelengths due to the strong wavelength dependence of atmospheric Rayleigh scattering (Hudson et al., 2006). Many satellite reflectance measurements are hemispherical-conical because they are the integrated signal of radiance upwelling from a naturally illuminated surface over some finite solid viewing angle (Grenfell, 2011) and therefore significant post-processing is often required to provide satellite albedo products. For conversion to albedo,

these directional or conical reflectance values must be integrated over the entire overlying hemisphere, taking surface anisotropy into account (see section 4.8).

Certain spectral conditions must also be met for a measured reflectance to be termed 'albedo'. Since broadband albedo is a crucial determinant of ice and snow energy balance, it is essential that measurements take into account a large fraction of the

incident solar irradiance. Solar irradiance extends well beyond the visible wavelengths, meaning opposed photosynthetically active radiation (PAR) sensors that typically only measure between about 0.4 and 0.7 µm are insufficient for measuring broadband albedo. Since UV and visible wavelengths account for only about half of the incoming solar energy (Painter et al. 2012; Liou, 2002, Table 2.3), studies that report values of bioalbedo using PAR sensors (e.g. Lutz et al., 2014, 2016) likely overestimate the albedo reducing effect of biological impurities because the influence of the impurities are restricted to the

visible and near-UV wavelengths. Over 99% of incident solar energy lies in the wavelength range 0.3 - 4 µm, which should be the standard range for characterizing the broadband albedo of a sunlit surface.

### 4.2: Characterizing snow or ice optical characteristics

Snow or ice physical characteristics are the primary driver of albedo (Fig.1B). This is because scattering occurs at ice-air interfaces. The greater area of interfaces per unit volume of snow/ice, the more likely a photon is to scatter back out of the snowpack before being absorbed (Warren, 1982). A high specific surface area (i.e. small effective grain size) is therefore associated with higher albedo, where specific surface area (SSA) is defined according to Equation 3 and is related to the effective radius ($r_{eff}$) by Equation 4 (all terms defined in Table 1).


$$SSA = \frac{S}{M} = \frac{S}{\rho_{ice}*V} \qquad \text{(Eq. 3)}$$

$$r_{eff} = \frac{3}{\rho_{ice} \times SSA} \qquad \text{(Eq. 4)}$$

For pure snow, grain size has negligible impact upon spectral reflectance at $0.3 - 0.45$ µm wavelengths where ice is virtually non-absorbing (Fig. 1B; Warren and Brandt, 2008), however it has a major impact at the NIR wavelengths. Infilling of interstitial spaces by meltwater replaces air-ice interfaces with water-ice interfaces and promotes forward scattering of light

deeper into the snow or ice, lowering the albedo (Gardner and Sharp, 2010). This is a key albedo-melt feedback that explains why wet snow has a lower albedo than dry snow (with the albedo reduction occurring mostly at NIR wavelengths). Radiative transfer schemes tend to parameterize the geometry of the ice grains using the SSA (or effective grain size) under the assumption that the real snowpack can be well described by a model snowpack of homogeneous spheres with the same total surface area and total mass as the real snowpack (Warren, 1982; Grenfell et al., 1994). However, determining these values empirically for real ice and snow is challenging. For snow, the integral of an absorption feature centred around 1.03 µm scales with grain size and can be measured using contact-spectroscopy (Fig. 1C; Nolin and Dozier, 2000; Painter et al., 2007). However, this may not hold for large grain radii or high impurity concentrations, and the absorption feature can be shifted to shorter wavelengths by liquid water (Green et al., 2002, 2006; Gallet et al. 2014). The same may be true for NIR (0.7 – 1.3 µm) photographic methods (Yamaguchi et al. 2014). Other options include microcomputed tomography (Micro-CT: Dadic et al., 2013) although this requires cores to remain completely frozen through extraction, transport and analysis. Grain sizes could be determined using a polarizing hand lens (Hubbard and Glasser, 2005; Negi et al., 2010) - although the error can be high (Painter et al., 2007) - or semi-automated analysis of photographs of thin/thick sections (Dadic et al., 2013) or the sample surface. Non-spherical grain shapes can influence the surface albedo (Dang et al, 2016) and can be simulated in the radiative transfer scheme TARTES (Libois et al., 2013). It is also important to note that optical grain size and observable grain size are different properties and high uncertainty results from assuming that they are equivalent – this is because the observable grain size does not represent the absorbing path length.

### 4.3: Characterizing the incoming solar irradiance

Broadband albedo is an apparent property of a surface that depends upon the spectral and angular distribution of incoming irradiance as well as the inherent optical properties of the surface itself. Incoming solar irradiance is modified by the solar zenith angle, atmospheric composition, cloud cover and cloud optical thickness (Gardner and Sharp, 2010). Clouds increase the diffuse portion of the incoming irradiance and preferentially absorb near-IR wavelengths and scatter near-UV and visible wavelengths upwelling from the surface, resulting in a spectral shift towards shorter wavelengths. Cloud cover therefore increases the broadband albedo of the surface relative to clear-sky conditions (Grenfell and Maykut, 1977; Grenfell and Perovich, 1984; Gardner and Sharp, 2010). Furthermore, scattering and absorption by atmospheric gases and aerosols modify the spectral quality and attenuate the top-of-atmosphere solar irradiance.

Incoming solar irradiance is best measured using a spectral radiometer with a cosine-collector oriented to look upwards (Grenfell, 1981). The cosine collector enables spectral measurements to be made over the entire overlying hemisphere, and can then be rotated to look downwards to detect upwelling irradiance, thereby enabling a surface albedo measurement. Sometimes, instruments with limited field of view (FOV) might be used (discussed in section 4.10). In this case, the spectral distribution of incoming solar irradiance is best measured using a white-reference panel (Schaepman-Strub et al., 2006; McCoy, 2005; Milton et al., 2009). Because the panel is as close as possible to an isotropic scattering surface, light reflected from it approximates the spectral characteristics of incoming solar irradiance. Common materials used for white reference panels are Spectralon™ and Barium Sulphate (Jaglarz et al., 2006). The white reference material used must approximate a Lambertian scattering surface as closely as possible so that anisotropic reflectance and spectral absorption effects are minimized in the measurement standard, although even Spectralon™ surfaces exhibit a degree of anisotropy (Bonnefoy, 2001; Voss and Chang, 2006). For occasions where field data are unavailable, incoming solar irradiance can be modelled using the latitude, longitude, time of day and site specific information or approximations regarding the atmospheric profile and an atmospheric radiative transfer model such as SBDART or COART. In our simulations, we use the default SNICAR incoming irradiance, but direct the reader to our README file that accompanies BioSNICAR where the protocol for updating the

incoming irradiance for specific sites using atmospheric RTMs is described. In areas of high relief a surface model that includes multiple reflections and site-specific radiative processes may be required.

### 4.4: Characterizing the optical properties of individual cells

The optical properties of individual cells are difficult to measure in the field for several reasons. Spectral measurements resolved at the scale of a single cell require equipment that cannot be readily utilized in the field. Yallop et al. (2012) successfully employed microspectrophotometry in the laboratory to identify some pigments in ice algae. However, the freezing, transportation, storage, thawing and filtering of the samples required for this analytical technique limit its utility to the more stable pigments. Furthermore, the concentrations of each pigment (specifically those associated with xanthophyll cycling) are sensitive to changes in the light field experienced during sampling (Demers et al., 1991; Grant and Louda, 2010). Changes in the light field can stimulate or relax non-photochemical quenching mechanisms and open or close photosynthetic reaction centres in the algal cells, leading to rapid changes to their pigment profiles (Grant and Louda, 2010). With knowledge of biomass concentration and distribution and the surrounding medium, an inverse modelling approach may allow pigment mass fractions to be estimated (e.g. Bigidare et al., 1990; Bricaud et al., 2004; Moisan et al., 2011) which could in turn enable inferences to be made about environmental stresses and stress-responses in the ice surface algal ecosystem.

The simulations shown in Figure 2 demonstrate that increased secondary carotenoid concentration results in lower broadband albedo but also changes the shape of the spectral reflectance curve. Carotenoids lower the albedo between $0.4 - 0.6$ µm and have the effect of masking the spectral signature of chlorophyll and extinguishing the characteristic 'chlorophyll bump' (e.g. Seager et al., 2005). This effect may hinder attempts to determine cell concentrations by remotely sensing chlorophyll absorption features, despite this being feasible for other algal blooms (e.g. lakes and oceans) and vegetation mapping. We simulated four pigment scenarios: high (1.5% chlorophyll a, 10% primary carotenoids, 10% secondary carotenoids), medium (1.5% chlorophyll a, 5% primary carotenoids, 5% secondary carotenoids), low (1.5% chlorophyll a, 1% primary carotenoids, 1% secondary carotenoids) and chlorophyll-only (1.5% chlorophyll a). We were guided by published values for algal pigment content, although they vary widely because pigmentation changes in response to environmental stresses and between species (e.g. Lamers et al., 2008). For example, Griffiths et al. (2011) found pigments to comprise $0.5 - 5\%$ of cellular dry weight in the microalga Chlorella vulgaris, Christaki et al. (2015) suggested between 0.5 - 8% for phycobiliproteins in algal cells and up to 14% for carotenes specifically, while 1-2% is commonly suggested for chlorophyll a (Christaki et al., 2015; Kirk et al., 1975). Secondary carotenoid content between 7.7 and 10% for secondary carotenoids was reported for light-stressed microalgae by Lamers et al. (2008). Each pigment has a unique effect on the cell optical properties. For our simulations the pigments were mixed according to Equation 1. In each scenario there was no interstitial water, the ice grains had a constant radius of 1500 µm and the biomass of the biological impurities in a 3 mm surface layer was varied (here, 0.01, 0.05, 0.1, 0.5, 1, 1.5 and 2 $mg^{alg}/g^{snow}$). Increasing the biomass from 0.01 to 2 $mg^{alg}/g^{snow}$ reduced the broadband albedo from 0.68 to 0.38 for the high carotenoid scenario, 0.39 for the medium carotenoid scenario, 0.41 for the low carotenoid scenario and 0.50 for the chlorophyll only scenario (Fig. 2). This highlights the importance of pigmentation for albedo reduction but also suggests that once a threshold concentration of carotenoid pigments is present in the cell further increases have diminishing impact on their albedo-reducing efficiency (see Appendix 2). The presence of carotenoids masks the spectral signature of the chlorophyll (Fig. 2). Previous empirical studies have reported biomass concentrations as high as $5.2 \times 10^4$ cells $mL^{-1}$ (Lutz et al., 2014) for melting snow in eastern Greenland and $29.5 \times 10^4$ cells $mL^{-1}$ for ice algae in western Greenland (Yallop et al., 2012) although it is not possible to convert this into $g^{algae}/g^{snow}$ without knowing precisely the dimensions and physical characteristics of the algal cells and snow/ice sampled.

The optical properties of algal cells also depend upon cell size. BioSNICAR calculates the Mie optical properties for algal cells of different radii before appending them to the lookup library. In Figure 3 we show results for small (radius = 5 µm), medium (radius = 15 µm) and large cells (radius = 25 µm). The model assumes the cells to be spherical and homogeneous; however, ice algal cells are often irregularly shaped. We apply an effective spherical radius approach (again a collection of spherical cells with equivalent SSAs, as was done for nonspherical snow grains) justified by the random orientation of diverse cells. Decreasing cell size for the same total biomass is shown to enhance the albedo reducing effect (the broadband albedo for the highest simulated biomass is 0.22 for small cells, 0.39 for medium cells and 0.48 for large cells). This albedo reduction is concentrated at wavelengths < 1.1 µm, whereas at wavelengths > 1.1 µm the albedo can actually increase with additional biomass, especially for smaller cells. These observations are consistent with inorganic impurities including soot and mineral dusts (Figs 4 and 5 in Warren and Wiscombe, 1980; Flanner et al., 2007) and results from the total cross section of many smaller particles being greater than fewer larger particles for an equal impurity mass. Skiles et al., 2017 and Skiles and Painter (2017) found that modelled albedo decline due to dust and black carbon in snow was dissimilar to measurements, indicating the urgent need for radiative transfer modelling of biological impurities to be field validated. We have recently undertaken field work for this purpose and will publish the outcomes in a future paper. There is uncertainty in the biological literature concerning the impact of cell size on the optical properties of algae arising from the spatial distribution of pigments within cells (Haardt and Maske, 1987), variable intracellular architecture (Haardt and Maske, 1987), packaging effects (Morel and Bricaud, 1981) and non-linear relationships between cell growth and pigment mass fraction (Alvarez et al., 2017). These simulations confirm that empirical determination of cell size is important for radiative transfer modelling of bioalbedo, and further emphasise the need for studies focussed on the links between photophysiology and optical properties of ice algae. We expect that the range of cell sizes modelled here represent a likely range for real ice algal cells, although ice-dwelling cyanobacteria can be filamentous and several tens of microns in length (Uetake et al., 2010; Yallop et al., 2012).

### 4.5: Measuring biomass

The biomass concentration of cells ($g^{algae}$/$g^{ice}$) in the snow or ice medium is a crucial determinant of surface albedo (Fig. 2, 3, 4). Impurity loading is required per unit volume or mass of ice or snow rather than per unit area of surface although the distribution of cells within the volume at the time of sampling is also crucial (see Section 4.6). Cells can be counted in field samples using traditional microscopy or flow cytometry, although the two methods may produce different results (Stibal et al., 2015). The chlorophyll content of a sample has traditionally been used as a proxy for biomass; however, the amount of chlorophyll in an individual cell is species-specific and changes over time as a mechanism of photoacclimation (e.g. Felip and Catalan, 2000) making the use of chlorophyll-based biomass estimates questionable, especially in bioalbedo studies where both the pigmentation and abundance of cells are crucial variables. Isolating the biological contribution to snow and ice albedo is crucial for modelling ice dynamics under future climate scenarios where the intensity and spatial coverage of algal blooms will likely vary differently to inorganic impurities. For snow, Painter et al. (2001) developed a method for retrieving snow algal biomass from spectral reflectance. This is not directly applicable to algal blooms on ice for several reasons: ice algae produce additional pigments that can obscure the diagnostic chlorophyll absorption feature used by Painter et al. (2001), and dust loading on ablating ice is often more prevalent than on seasonal snow. Heavy dust loading can also reduce reflectance in the visible wavelengths in such a way as to be indistinguishable from the chlorophyll-based 'red-edge' biosignature (Fig. 4C) that is commonly used to remotely detect terrestrial vegetation and ocean plankton blooms (e.g. Roling et al., 2015; Joint and Groom, 2000). Therefore, the red-edge biomarker can only be used to remotely sense ice algae if dust can be accurately accounted for, otherwise there is a high risk of biomarker false positive. In addition, melting glacier surfaces generally have large surface roughness features (e.g. Smeets and van den Broecke, 2008) and less efficient scattering, both of which increase the anisotropy of scattering relative to snow. On the other hand, rough surfaces can create photon cavities that increase trapping

and absorption, lowering albedo (Cathles et al., 2011). Increased surface and shallow-subsurface heterogeneity, combined with greater inorganic impurity loading make biological signatures more difficult to distinguish from spectral noise. The chlorophyll absorption feature, already subtle due to the presence of additional pigmentation in ice algae compared to snow algae can be further attenuated by this noise.

### 4.6: Characterizing the depth-distribution of cells

The vertical distribution of cells within snow or ice is an important factor in determining their impact on albedo (Fig. 4A). This has an important influence upon biomass measurement. When field studies report number of cells per unit volume of melted snow or ice, the depth from which the volume of ice was removed must also be reported. For two samples of equal volume from ice with biological impurities concentrated into the upper surface, sampling very shallow ice over a wider area will result in a high cell concentration compared to a narrow, deep sample where more clean subsurface ice is included. Ice algae concentrate into a very thin surface layer of the order of 1 mm (Lutz et al., 2014; Yallop et al., 2012). Best practice would involve removing samples at well-defined depth intervals. There is a practical limit of approximately 2 cm for removing weathered ice, because it is often a rough surface. However, careful visual inspection or photographic records may provide further detail about vertical distribution at spatial resolution < 2cm to support modelling. As shown in Figure 4A, the albedo impact of distributing the same biomass concentration over 1 mm and 1 cm is dramatic.

### 4.7: Characterizing abiotic impurities

Inorganic light-absorbing impurities include mineral dusts and black carbon. These impurities have an albedo reducing effect (Warren and Wiscombe, 1980; Warren, 1984; Gardner and Sharp, 2010) and can be difficult to distinguish from biological albedo reduction, having similar spectral absorption signatures in the visible wavelengths (Fig. 4B, C). Isolating the inorganic impurity loading is therefore necessary for quantifying bioalbedo. By incorporating measured inorganic impurities into a radiative transfer scheme, and knowing the ice properties, the effects of biological impurities can be determined to first order by differencing the real spectrum from a modelled inorganic impurity-only spectra (although where impurities are mixed vertically more complex non-linear unmixing techniques are required). To quantify inorganic particulates, ice samples can be melted and filtered onto quartz fibre filters and total inorganic mass measured using thermogravimetry, which can be combined with X-ray diffraction analysis to characterize the composition of the minerals (e.g. Smith et al., 2016). Scanning electron microscopy with energy dispersive x-ray spectroscopy (SEM-EDX) can provide information on the geochemistry of samples and particle size distributions. There are also many optical microscopy techniques that can be used to determine sample mineralogy (Winchell, 1951; Shelley, 1985). It is not unreasonable to assume that the mineralogy of cryoconite is representative of the surface ice, since cryoconite acts as a store for mineral grains from all sources in the supraglacial zone (Stibal et al. 2012; Cook et al. 2016). Therefore, the same analyses can be applied to cryoconite grains with the benefit of easy sample collection and no need to filter onto papers, although the inorganic components must be extracted from the organic matter chemically or by combustion. Dust can also be delivered to the surface by melt-out from underlying glacial ice. Dust loading on ice obscures the spectral signature of algal cells in the visible to NIR wavelengths (Fig. 4C). While the effect of dusts may be negligible in many cases, the possibility of very high concentrations in some areas (e.g. Wientjes et al., 2011) and the similarity of spectral signatures resulting from biotic and abiotic impurities make abiotic impurities a crucial component of bioalbedo investigations, especially those aiming to detect biological impurities remotely or quantifying biological albedo reduction empirically. The reflectance of some minerals is difficult to distinguish from that of microbes (Seager et al., 2005) and interactions of microbes with minerals can obscure or modify biological reflectance spectra (Roling et al., 2015). Therefore, if a chlorophyll or carotenoid based biomarker is designed to be universally applicable (e.g. detection

of photosynthetic life across Earth's ice sheets and glaciers as well as extraterrestrial ice) then it must distinguish biotic from abiotic impurities. Our modelling shows that for equal mass concentrations, algal cells are generally more effective albedo reducers than mineral dusts (Figs 2 and 4) although this depends upon the specific optical properties of the impurities. This was found to be true for three other classes of dust (size fractions 0.05-0.5 um, 0.5-1.25um, 1.25-2.5 um) although the data are not shown here. Real dust optical properties vary widely and SNICAR uses a global mean that is unlikely to be a good representation of real dusts in any specific location. In our simulations black carbon was found to be a more effective albedo reducer than algal cells per unit mass (data not shown).

### 4.8: Accounting for surface anisotropy

Anisotropic scattering, characterized by the BRDF, causes reflectance measured over a particular solid angle to differ from the hemispheric albedo. This occurs because ice scatters light preferentially in the forward direction (Winther, 1993). The degree of anisotropy may exhibit significant spatiotemporal variability since enhanced forward scattering is generally associated with smoother surfaces and enhanced backscattering is generally associated with rougher surfaces (Nolin and Payne, 2007) and the scattering direction is dependent upon the directionality of incoming irradiance. The correction for this is known as the anisotropic reflectance factor (ARF) which is derived from empirical angular reflectance measurements approximating to the bidirectional reflectance distribution function (Hudson et al., 2006). The ARF varies according to the optical properties of the medium and impurity content (Warren, 2013). Several studies have estimated BRDFs and HCRFs of snow and sea ice surfaces (e.g. Hudson et al., (2006); Marks et al., (2015); Arnold et al., (2002), but equivalent measurements are scarce for glacier ice in various states of melt and impurity loading, although Naegeli et al., (2015) measured HCRF for various surface types on an alpine glacier and Gruell and de Ruyter de Wildt (1999) measured anisotropic reflection from melting glaciers in two Landsat TM bands. Naegeli et al., (2017) recently showed that ice surfaces with higher impurity loads scatter more anisotropically, possibly due to locally enhanced melt. ARF's for ice and snow containing biological impurities have not yet been reported and the ARF's may change depending upon cell size, shape and pigment content.

### 4.9: Accounting for indirect biological albedo feedback

In addition to the direct effects of algae on lowering ice albedo there are also indirect effects such as increased meltwater generation and modified ice-grain evolution. A bioalbedo-induced increase in the net radiation balance at the surface can change the rate of ice grain evolution and produce meltwater that fills interstitial pore spaces or coats ice grains. All of these indirect effects tend to reduce the number of scatters near the surface, reducing the albedo across the solar spectrum (Fig. 1A, Fig4D). These same effects are true of abiotic impurities. Biological impurities on ice are likely to be algal or cyanobacterial cells which can be prone to attachment to mineral fragments and/or forming biofilms. This may capture mineral fragments and prevent their removal by meltwater, thereby increasing the total abiotic impurity loading on an ice or snow surface. Because of the complex interplay between these processes direct and indirect quantification of the effect of biological impurities on ice is challenging to quantify.

### 4.10: Standardizing measurement and instrument configurations

To measure the albedo of a surface, a cosine collector can be used (Grenfell, 1981). This should be fixed in position perpendicular to the surface and rotated through 180° to look upwards and downwards (Fig. 5). Working under overcast skies allows for the quantification of albedo under diffuse illumination. To avoid error resulting from variable cloud conditions, downwards irradiance measurements can be made before and after upwards irradiance measurements and interpolated to the

precise measurement time to calculate the albedo (S. Warren, personal communication). Depending upon the frame used to support the sensor, a shadowing correction should be applied (Nicolaus et al., 2010; For reference, Brandt et al., 2011 used a 1.7% shadowing correction). The user should maintain sufficient distance from the sensor to avoid shading or reflection from clothing.

For characterizing small sample surfaces on heterogeneous ablating ice (e.g. measuring the albedo of an algal bloom) the cosine-collector method suffers from several limitations. Ablating ice is often highly heterogeneous at scales of centimetres to metres meaning that albedo measurements made using cosine collectors with wide viewing-angles will necessarily be the integrated signal from a variety of surfaces. This can be mitigated to some extent by reducing the height of the sensor, but to determine the reflectance properties of a surface component with dimensions of the order of decimetres (e.g. an algal bloom), a hemispheric-conical reflectance measurements (HCR: Fig. 1A) made using a sensor with a limited field of view are likely more appropriate and can be used to estimate the albedo by integrating over the entire viewing hemisphere, accounting for anisotropic scattering. The spectral radiometer fibre optic is collimated using a lens with a field-of-view that restricts the ground viewing area. To achieve this, measurements are made from a constant viewing angle (usually nadir) at a fixed distance above a sample surface (Fig. 5). The sensor alternately measures light reflected and scattered from the sample surface and a white reference panel. With knowledge of ARFs for the sample surface and reflectance panel, these measurements can be used in conjunction with an angular integration model to approximate albedo (Grenfell, 2011; Casey et al., 2016). This measurement also requires correction for instrument shading (Nicolaus et al., 2010; Grenfell, 2011). Finally, for reflectance-factor measurements the white reference panel must be clean and flat (Schaepman-Strub et al., 2006; McCoy et al., 2005; Milton et al., 2009) and a white reference measurement should be taken before and after each new measurement so that changes in solar angle, cloud cover or atmospheric effects can be accounted for. The variation between albedo measured using a cosine collector and reflectance measured using an 8° collimating lens at nadir view for the same surfaces (clean ice and algal ice near the IMAU Weather station 'S6' on the southwest Greenland Ice Sheet) is demonstrated in Figure 6. The measurements were all made within 15 minutes during constant clear sky conditions on 14th July 2017 using the same spectral radiometer, fibre optic and tripod arrangement that did not move between measurements. The variation is most likely due to spatial integration of a wider range of surfaces for the cosine collector and the lack of angular integration for the collimating lens.

For both albedo and HCR measurements, there are additional instrument and procedural considerations. Prior to use, the spectral radiometer should be allowed time to initiate, due to the different warm-up rates of three internal spectral radiometer arrays. Failure to do so can introduce 'step' artefacts into the measured spectra. For the commonly used ASD FieldSpec Pro (Analytical Spectral Devices), optimum warm up times of 90 minutes are recommended (NERC Field Spectroscopy Facility, 2007); however, this may not be feasible in the field when battery power in low temperatures is an issue. In these cases it is possible to warm up the spectral radiometer in advance using external power or a 'sacrificial' battery before switching to a new battery for deployment in the field. The instrument set-up also involves selecting the most appropriate fore-optic. For measurements integrating over a given area of sample surface the height the sensor should be held can be calculated using the instantaneous field of view (IFOV) of the fore-optic. Since there are uncertainties related to the true IFOV of a spectral radiometer, even when collimated with a fore-optic (MacArthur et al., 2012), the sample surface should be a) approximately homogeneous, and b) significantly larger than the area observed by the spectral radiometer. Some degree of spatial integration is desirable for many bioalbedo studies. To avoid saturating the sensors under changing illumination, the spectral radiometer must also be optimized before each measurement to allow the sensors to adjust to new lighting conditions.

Ideally, measurements should be taken within 2 hours of solar noon to minimize changes in illumination due to solar angle. The measurement itself must be made at a constant angle (typically nadir). This can be achieved using a bubble-level on a

pistol-grip (e.g. ASD's pistol-grip) or by setting the sensor on a tripod. Sensor tilt can introduce significant measurement error. Errors due to surface slope are greatest under clear skies with large solar zenith angles and can be minimised by making albedo measurements close to solar noon and corrected for in post-processing (Grenfell et al., 1994). Several pseudo-replicate measurements should be averaged to minimize the instrument error for a particular measurement. Unless specific angular

measurements are being made, the user and instrument should be oriented towards the incoming solar irradiance to prevent shading the sample surface. For more specific information regarding instrument and measurement configurations for field spectroscopy, we suggest the NERC FSF instrument guide (http://fsf.nerc.ac.uk/resources/guides/) the ASD User Guide (http://support.asdi.com/Document/FileGet.aspx?f=600000.PDF), a critique of field spectroscopy by MacArthur et al. (2015) and the overview provided by Grenfell (2011).

For some studies, a measure of broadband - rather than spectral - albedo is sufficient. In this case the measurement is more straightforward. Opposed pyranometers have commonly been used to measure broadband albedo (e.g. Cutler and Munro, 1992; Yallop et al., 2012; Lutz et al., 2014) with the ratio of irradiance measured by the downwards-looking and upwards-looking pyranometers providing a measure of surface albedo. This requires that each pyranometer has a viewing geometry of 180°

zenith over 360° azimuth (i.e. hemispheric) and that the spectral range of the pyranometer is at least 300-2500 nm, representing 95% of the total incoming solar energy (Kopp et al., 2005) (see Section 4.1). Tilt error is very important, so some measure of tilt angle (e.g. bubble level) is crucial for the opposed pyranometer method. The pyranometers must also be held a sufficient distance from the user or fixed to a tripod with a lever-arm to avoid shading or reflection from the operator. A consistent height above the surface must also be maintained to ensure the same size of sampling area between measurements. The guideline that

measurements be taken within 2 hours of solar noon also applies. Since the change in solar angle over time will vary seasonally and with latitude and the acceptable range of solar zenith angles will vary between experiments, the best approach is to identify an appropriate sampling window for each specific study. Opposed pyranometers allow albedo measurements to be determined directly but requires the two sensors to be cross-calibrated. Using a single pyranometer and rotating upwards and downwards negates the need for cross calibration.

For albedo measurements by any method, metadata collection is crucial. The minimum required metadata can be divided into three categories: a) instrument configuration; b) illumination conditions; c) surface conditions. This metadata is specific to each measurement and is required for transparency and appropriate interpretation of the data. We provide data booking sheets for bioalbedo studies in Appendix 3. We suggest that these booking sheets, or versions thereof, can be used to standardize

bioalbedo measurements for the purpose of integrating spectral reflectance, glaciological and biological measurements, and facilitate effective communication between bioalbedo modellers and empiricists.

**5 Conclusion:**

Bioalbedo is a significant component of the energy balance of glaciers and ice sheets that is yet to be quantified. The lack of understanding of the physical mechanisms driving albedo reduction by biological impurities and difficulty in isolating the biological albedo signal from inorganic impurities and ice optical properties remain significant hurdles for bioalbedo research. In this paper, a new physical model was presented, providing a framework for studying biological albedo reduction from first principles. The model was used to quantify the effects of biological variables such as biomass, distribution in an ice column,

pigmentation, and cell size on the surface albedo. Biological impurities were confirmed to be potentially significant components of the surface albedo, and the optical properties of the cells were found to be crucial determinants of the magnitude of their albedo lowering effect. However, it was emphasised that empirical knowledge of bioalbedo is currently lacking. Ten

specific challenges to integrating theoretical and empirical studies or linking ground measurements to remotely sensed spectral data were identified and quantified using the model, leading to suggestions for improved protocols for field studies.

**6. Code Availability**

The code and all dependencies are available in our repository (https://bitbucket.org/jmcook/biosnicar) along with an instruction manual and README.

**8 Data availability**

The data used in our modelling experiments are provided in our repository (https://bitbucket.org/jmcook/biosnicar).

**9 Author contributions**

JC developed the code, ran the experiments, wrote the paper and produced the figures. AJH, AG, MF, AT, CW, RB, JN, TDI and MT all provided useful comments and guidance regarding the content of the paper. AG and MF provided specific guidance on the model development and implementation. All authors refined manuscript drafts.

**10 Competing interests**

The authors cite no conflict of interest

**11 Disclaimer**

N/A

**12 Acknowledgements**

JC, AT, TIF, AH, MT, JN, CW and AG acknowledge UK-funded Natural Environment Research Council Consortium Grant
"Black and Bloom" (NE/M021025/1). JC acknowledges the Rolex Awards for Enterprise. AG and JN's contributions were supported by funding from the NASA Cryosphere programs. The original SNICAR codes were written by Mark Flanner (see Flanner et al., 2007). Steve Warren (University of Washington) is sincerely thanked for valuable comments on manuscript drafts and advice on field measurements.

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

**Figures:**

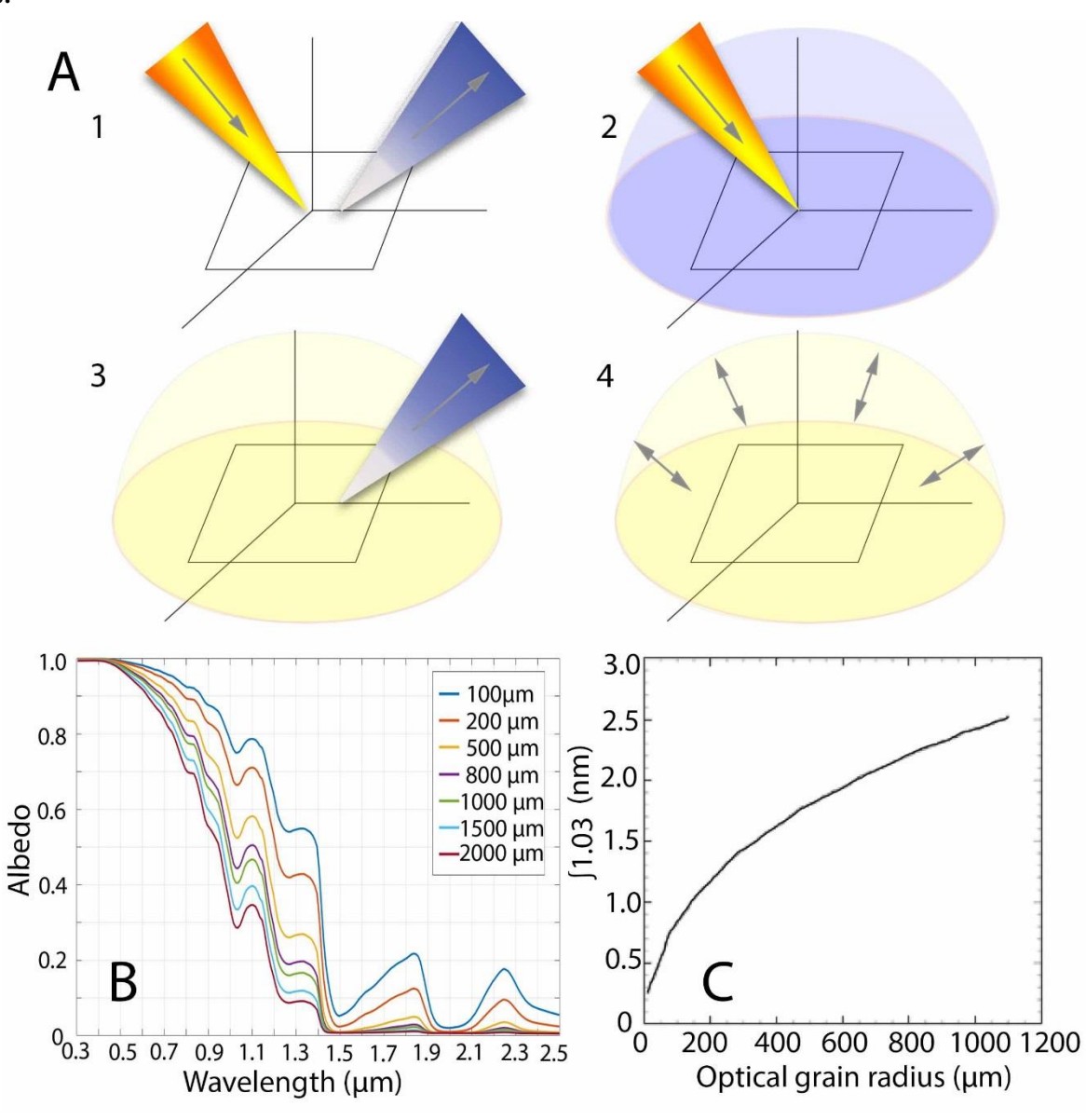

**Figure 1:** A) Diagram of the four measurable reflectance scenarios described by Nicodemus et al. (1977). 1 = biconical reflectance, 2 = conical hemispheric reflectance (blue hemisphere represents outgoing radiance), 3 = hemispheric-conical reflectance (yellow hemisphere represents incoming radiance), 4 = bihemispheric reflectance. B) Spectral albedo for clean snow with grain radii 100 - 2000 µm, solar zenith = 60°, no impurities; C) Relationship between optical grain radius and the integral of the 1.03 µm absorption feature, redrawn from Painter et al. (2007; Journal of Glaciology).

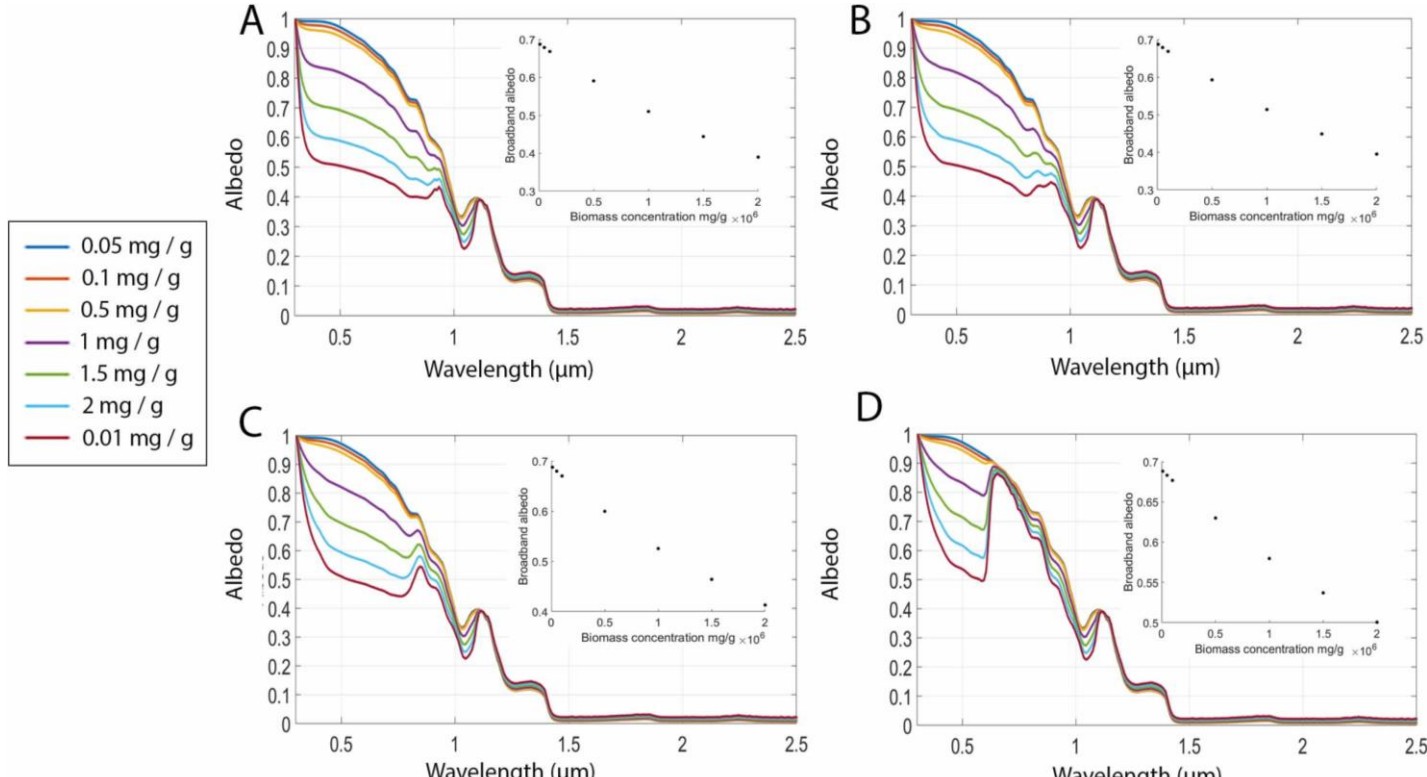

**Figure 2:** Spectral albedo of snow (grain radius 1500 µm) with equal biomass loading of algal cells with varying pigmentation. In all four simulations, chlorophyll a = 1.5% total cell dry weight. In A) Primary and Secondary carotenoids = 10% dry weight each. In B) primary and second carotenoids = 5% dry weight each. In C) primary and secondary carotenoids = 1% dry weight each. In D) no carotenoids are present, the cell contains chlorophyll only. In all simulations the solar zenith was 60°. The legend applies to all four subplots. Inset plots show broadband albedo against biomass concentration for each pigment mixture.

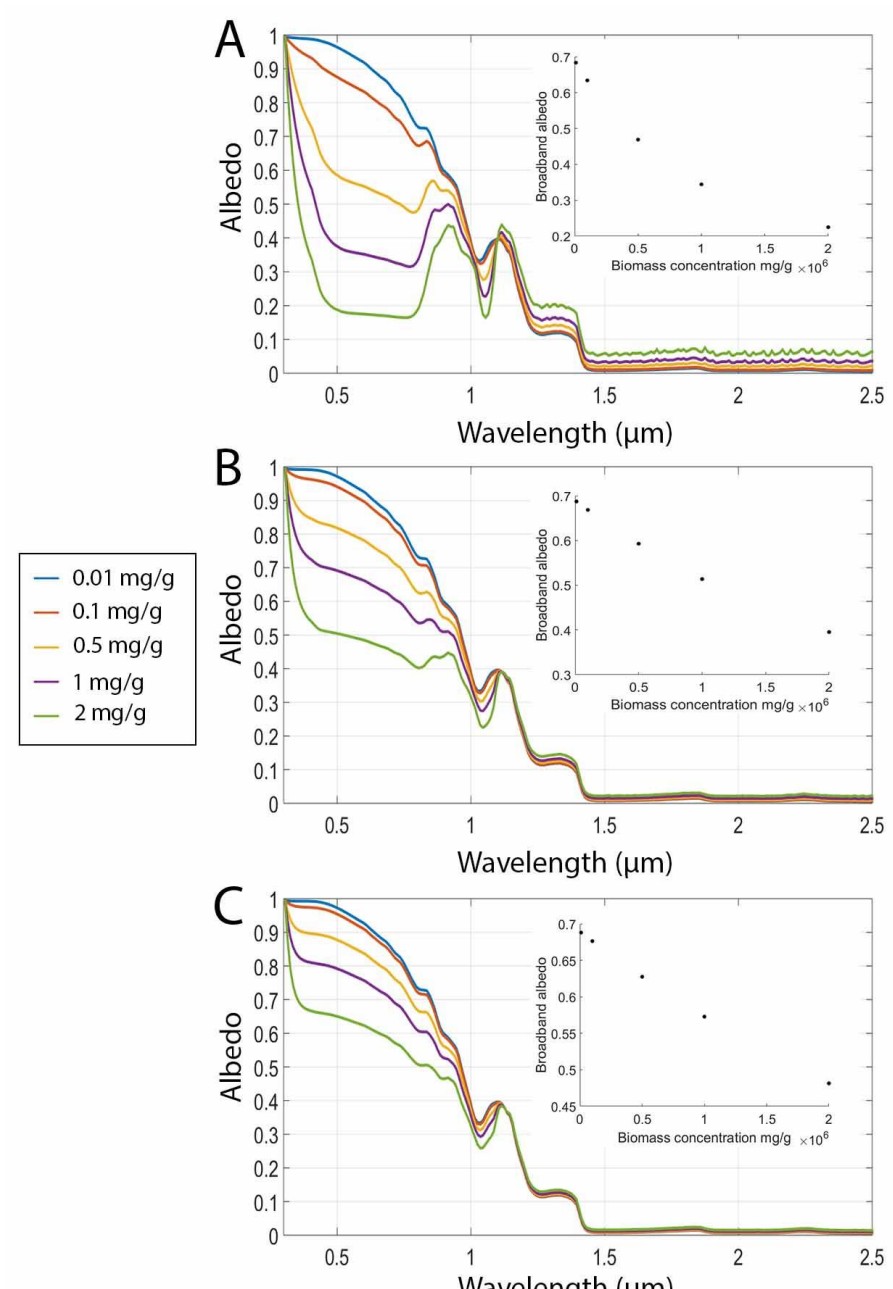

**Figure 3:** Simulations of 1500 µm radius ice grains with no interstitial water or inorganic impurities and biomass concentrations 0.01, 0.1, 0.5, 1 and 2 mg$^{alg}$/g$^{snow}$ confined to a thin (3 mm) surface layer. The mass fraction (% dry weight) of pigments in the cells was 1.5% for chlorophyll a and 5% for each of primary and secondary carotenoids. In A) the cell radius was 5 µm, in B) the cell radius was 15 µm and in C) the cell radius was 25 µm. In all plots the solar zenith was 60°. Legend applies to all three subplots. Insets show broadband albedo against biomass concentration for each cell size.

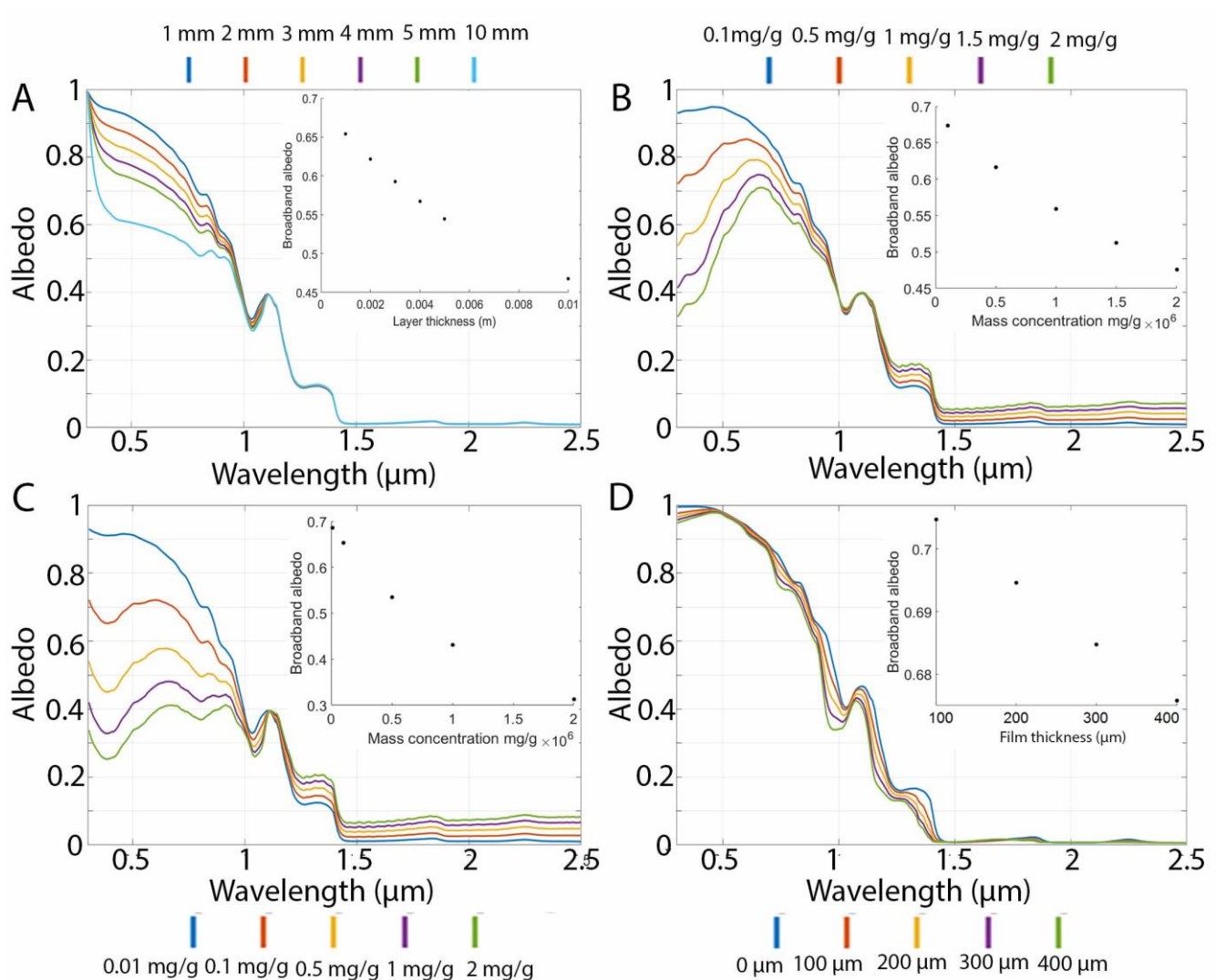

**Figure 4:** A constant biomass (0.5 mg$^{algae}$/ g$^{ice}$, pigment mass fractions (% total cell dry mass) = 1.5% chlorophyll a, 5% primary and secondary carotenoids, 15µm cell radius) distributed vertically in layers of ice (1500 µm grain radius) of varying thickness (1, 2, 3, 4, 5, 10 mm). B) Varying concentrations of mineral dust in a 3 mm surface layer (0.1, 0.5, 1, 1.5, 2 mg$^{dust}$/ g$^{ice}$) on otherwise clean ice (grain radius 1500 µm). The dust used was SNICAR's 'dust 4' which has grain radii 2.5 − 5 µm; C) Equal mass concentrations (0.01, 0.1, 0.5, 1, 2 mg$^{impurity}$/ g$^{ice}$) of algal cells (pigment mass fractions (% total cell dry mass) = 1.5% chlorophyll a, 5% primary and secondary carotenoids, 15 µm cell radius) and mineral dust (SNICAR's 'dust 4' which has grain radii 2.5 - 5 µm) in a 3 mm surface layer in otherwise clean ice (1500 µm grain radius); D) Albedo of a dry snowpack (grain radii = 1000 µm) and snowpacks with liquid water as a coating around the ice grains. The legend indicates the thickness of water layer around a 1000 µm ice grain. Insets show broadband albedo plotted against the relevant model variable.

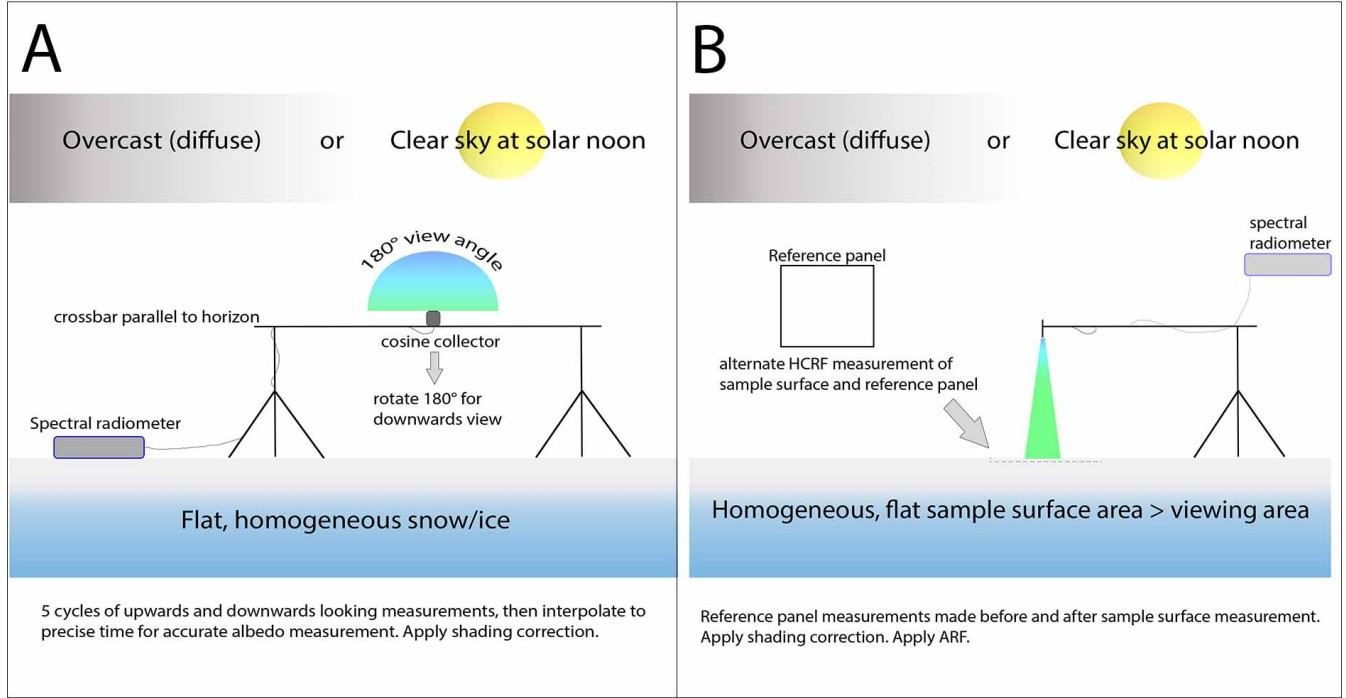

**Figure 5:** Schematic diagram of albedo measurement configurations for A) cosine collector, and B) HCRF measurement modes.

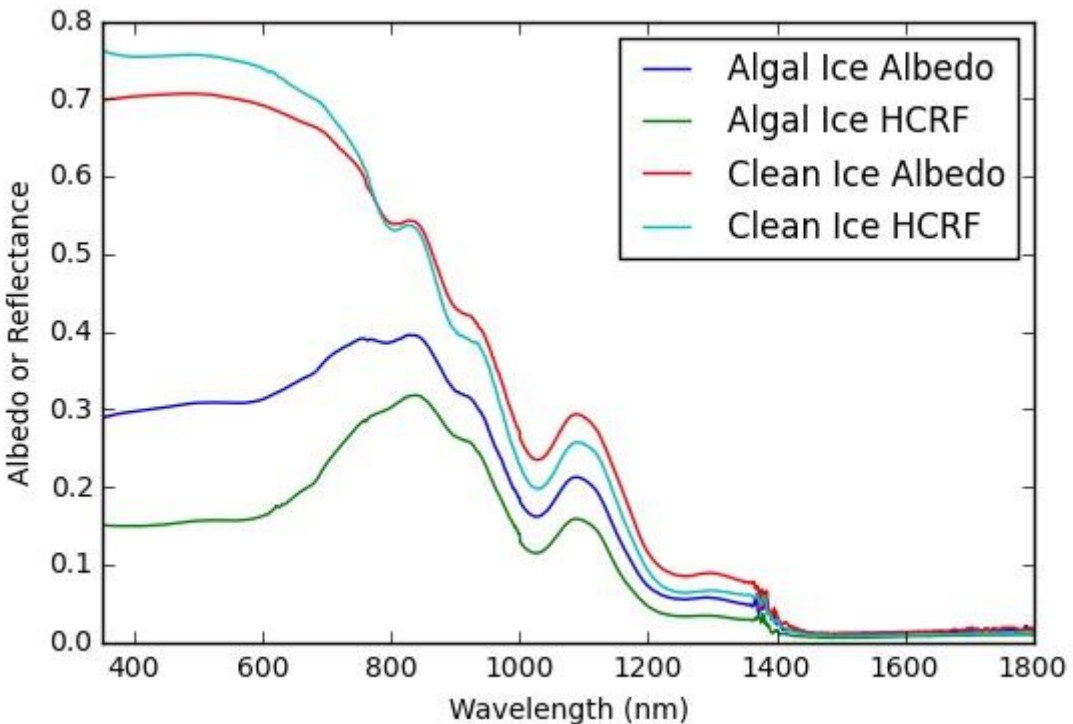

**Figure 6:** Field measured albedo (cosine collector) and nadir-view HCRF (8° collimating lens) for one clean ice and one algal ice surface. The spectra were collected using an ASD FieldSpec Pro 3 spectral radiometer following the methods described in Figure 5 and section 4.10 on 14th July 2017 during constant clear sky conditions near the IMAU Automatic Weather Station 'S6' on the Greenland Ice Sheet.

**Appendices**

5 **A1: Lookup Library for Biological Cells**

For all cells:

    Real Part of Refractive Index = 1.5

    Water fraction = 0.8

10   Density of dry cell = 1400 kg m$^{-3}$

| Name | Size (μm) | Chlorophyll a | Chlorophyll b | 1 Carotenoids | 2 Carotenoids | Phycocyanin | Phycoerythrin |
|------|------|------|------|------|------|------|------|
| Biological Impurity 1 | 30 | 1.5 | 0 | 10 | 10 | 0 | 0 |
| Biological Impurity 2 | 30 | 1.5 | 0 | 5 | 5 | 0 | 0 |
| Biological Impurity 3 | 30 | 1.5 | 0 | 1 | 1 | 0 | 0 |
| Biological Impurity 4 | 30 | 1.5 | 0 | 0 | 0 | 0 | 0 |
| Biological Impurity 5 | 10 | 1.5 | 0 | 5 | 5 | 0 | 0 |
| Biological Impurity 6 | 50 | 1.5 | 0 | 5 | 5 | 0 | 0 |

Table A1: Overview of cell size and pigment mass fractions (% total cell dry weight) for each biological impurity

**A2 Figure: Broadband albedo against pigment scenario**

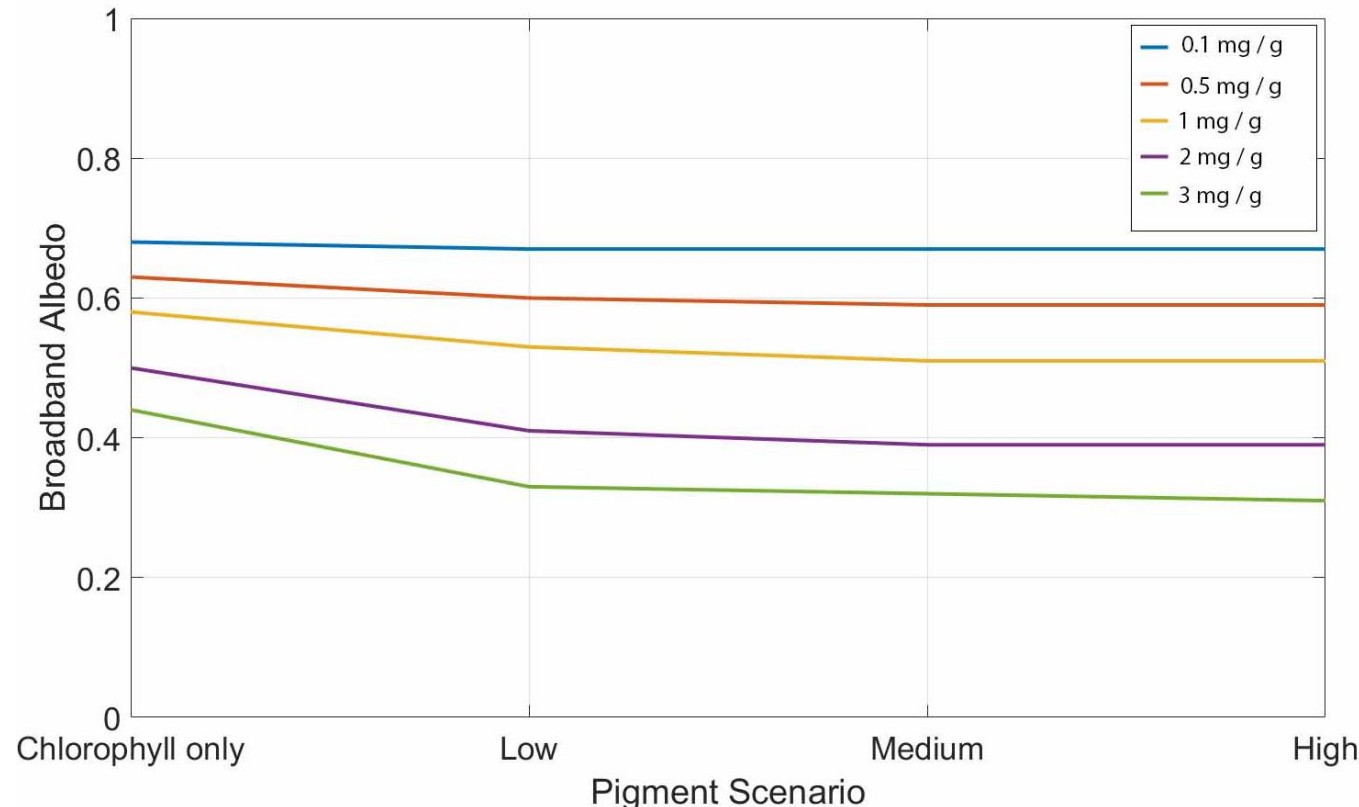

**Figure A2:** Broadband albedo against pigment scenario. In all simulations the ice crystal size was 1500 µm, the solar zenith angle was 60° and the impurities were confined to a 3mm surface layer. There was no dust or black carbon. The biological impurities had cell radii = 15 µm. The pigment scenarios are as follows:

Chlorophyll only: 1.5% chlorophyll a
Low: 1.5% Chlorophyll a, 1% secondary carotenoids
Medium: 1.5% chlorophyll a, 5 % secondary carotenoids
High: 1.5% chlorophyll a, 10% secondary carotenoids

The legend shows the biomass concentration of algal cells in the upper 3 mm layer.

# Bio-albedo Logsheet (V0.1): ASD Field Spec

5    Standardized metadata for pairing spectral reflectance with biological and glaciological measurements on ice surfaces. Designed for use with the ASD Field Spec with naturally illuminated sample surface. A new sheet with new general conditions will be required for each new site or new procedure.

| General Conditions | | | |
|---|---|---|---|
| Site (name, lat, lon, elev): | | Illumination Type: | |
| Date: | | Filename: Root | |
| Operator: | | Sample surface area (m²) | |
| Instrument On time: | | Sensor height (m) | |
| Reference Panel: | | Bio/dust sample dimensions (x,y,z: cm) | |
| Fore-optic | | Bio/dust sample volume: (cm³) | |
| Fo-jumper | | Solar zenith angle (°) | |
| Mode | | Sensor footprint area (m²) | |

| Time | Opt | DC | WR | Filename (root\...) | Description | Cloud (1/8, solar disc obscured?) | Surf Type (inc. local relief, slope angle) | Grain Size (mm) | Viewing angle (°) | Surf Img # | Bio/dust Smpl # |
|---|---|---|---|---|---|---|---|---|---|---|---|
| | | | | | | | | | | | |
| | | | | | | | | | | | |
| | | | | | | | | | | | |
| | | | | | | | | | | | |
| | | | | | | | | | | | |
| | | | | | | | | | | | |
| | | | | | | | | | | | |

# Bio-albedo Logsheet (V0.1): Pyranometry

Standardized metadata for pairing albedo measurements from pyranometery with biological and glaciological measurements on ice surfaces. Designed for use with naturally illuminated sample surfaces. A new sheet with new general conditions will be required for each new site or new procedure.

| General Conditions | | | |
|---|---|---|---|
| Site (name, lat, lon, elev): | | Sample surface FOV (m$^2$) | |
| Date: | | Sensor height above surface (m) | |
| Operator: | | Solar zenith angle  (°) | |
| Pyranometer model: | | Bio/dust sample dimensions (volume and depth interval) | |
| Illumination Type: | | Bio/dust sample volume: (cm$^3$) | |
| FOV (°) | | Filename: Root | |

| Time (GMT or Local?) | Filename (root\...) | Description | Cloud (1/8, solar disc obscured?) | Surf Type (inc. local relief, slope angle) | Grain Size (mm) | Tilt angle (°) | Surf Img # | Bio/dust Smpl # |
|---|---|---|---|---|---|---|---|---|
| | | | | | | | | |
| | | | | | | | | |
| | | | | | | | | |
| | | | | | | | | |
| | | | | | | | | |
| | | | | | | | | |
| | | | | | | | | |
| | | | | | | | | |

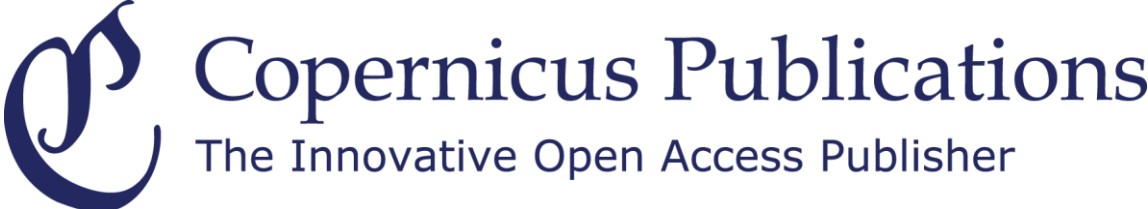

**Figure 1: The logo of Copernicus Publications.**