# Peer review of "Quantifying bioalbedo: A new physically-based model and discussion of empirical methods for characterizing biological influence on ice and snow albedo"

_The Cryosphere, 2017_

## Short Comment (SC1) · 30 May 2017

The authors developed a new snow/ice albedo model (Bio-SNICAR) to investigate the contamination effect of biological impurities in snow/ice. This is a powerful tool which can be used to advance our understanding on biological influence on snow/ice albedo. I have a short comment.

In radiative transfer modeling, the authors assumed spherical ice crystals and external impurity-ice/snow mixing (if I understand correctly). However, recent studies have found that ice/snow grain shape (Liou et al., 2014; Dang et al., 2016), ice/snow grain

packing/aggregating (He et al., 2017), and impurity-ice/snow internal mixing (Liou et al., 2014; He et al., 2017) have substantial impacts on albedo of clean and/or dirty ice/snow. Therefore, all these factors could introduce uncertainty in estimates of impurity effects on snow/ice albedo in the present manuscript. Could the authors add more details in model descriptions regarding how they dealt with these issues and include some additional discussions on these recent studies and issues?

References:

Dang, C., Q. Fu, and S. Warren, 2016: Effect of Snow Grain Shape on Snow Albedo, J. Atmos. Sci., 73, 3573–3583, doi: 10.1175/JAS-D-15-0276.1.

He, C., Y. Takano, and K. N. Liou, 2017: Close packing effects on clean and dirty snow albedo and associated climatic implications, Geophys. Res. Lett., 44, doi:10.1002/2017GL072916.

Liou, K. N., Y. Takano, C. He, P. Yang, R. L. Leung, Y. Gu, and W. L. Lee, 2014: Stochastic parameterization for light absorption by internally mixed BC/dust in snow grains for application to climate models, J. Geophys. Res.-Atmos., 119, 7616–7632, doi:10.1002/2014JD021665.

---

## Author Comment (AC1) · 2 Jun 2017

Dear Dr He,

Thank you very much for joining the discussion and for your supportive comments. You have understood correctly that the model is based upon the so-called "equivalent sphere" assumption and that the impurities are externally mixed. We are aware of the recent literature on grain shape influencing albedo and agree that incorporating these effects into the model would be a useful development aim. We have made the code

openly available to encourage other researchers to be active in enhancing the model and we have now added a comment to our README that highlights this opportunity to potential developers. However, we also point out that the incorporation of biological impurities into SNICAR represents a significant advance in our ability to quantify their effects on ice and snow albedo. To establish a framework for bioalbedo modelling it is appropriate to start with SNICAR because it remains the 'industry standard' for snow and ice albedo prediction. While the main aim of the paper is firstly to advance our bioalbedo modelling capabilities, it is also to establish standard operating procedures for measuring biological albedo reduction in the field. This is achieved using our current version of BioSNICAR. We agree that incorporating the literature you suggested would improve the paper, so we will expand our discussion of the effects of grain shape on snow albedo in our final manuscript.

Many thanks again for your input.

---

## Referee Comment (RC1) · Anonymous Referee #1 · 11 Jun 2017

This crystal clear, relevant, and very complete paper provides a physically-based framework for quantifying the effect of biological activity on the albedo of snow and ice. For a radiative transfer expert, this paper dwells quite long on RT theory and observational techniques. But I guess that the authors have a much broader audience in mind for this paper (but this is nowhere stated explicitly!), including biologists that have not worked much with RT before. In that sense, this manuscript is a great one-stop reference for everyone working on bioalbedo. I have only one major request: For figures 2, 3, and 4, could an additional panel be added showing the dependence of

the broadband albedo on the variable of interest? Otherwise, I only have some minor suggestions for improvement.

Note to the editor: my background is in radiative transfer modelling, so separate advice on the biological aspect of this paper should be sought by another reviewer.

P1L28-31 : I would cast some of the 10 challenges differently, such that it is clear what the challenge exactly is. E.g. "Ambiguity in terminology: -> "Reconciling ambiguous terminology"; "Surface anisotropy" -> "Accounting for surface anisotropy"; "Measurement and instrument configurations" -> "Standardizing measurement and instrument configurations" or similar.

P1L37: ... on THE ice surface ...

P4L19: You use both BioSNICAR and Bio-SNICAR (with and without hyphen). Please standardize.

P4L21: Please turn this list into a table, with an additional column showing the mathematical symbol used in this paper. Instead of the somewhat cumbersome lines 42-45.

P4L21: I guess that these library files are wavelength-dependent? At least, the items 2,3 and 4 in the list? Could you indicate which information is wavelength-dependent?

P5L39: This is a good example of a section that I would recommend to shorten significanly, if the audience of this paper were strictly limited to RT specialists. No new insights are presented here, and a paper like Schaepman-Strub et al. covers this entire section.

P7L15: strictly speaking, the rightmost part of the equation is a definition of r_eff. You could add a second equation here, defining r_eff as

r_eff = 3 / (rho_ice * SSA)

P10L11: Ambiguous. Please reformulate: "... unless dust can be accounted for accurately. Otherwise, there is a high risk of biomarker false positive"

[Figure]

P10L12: on the other hand, rough surfaces create photon cavities that increase scattering and absorption, and lower albedo (e.g., Cathles et al., 2011, Ann. Glac. 52(59))

P13L14: straightforward

Figure 2: It would be very illustrative to add a panel here that shows broadband albedo as a function of pigmentation level. I understand that this requires an additional setting, namely the prescription of an atmospheric vertical profile determining the spectral composition of the radiation arriving at the surface. Nonetheless, I believe that this would be very instructive. How much does the presence of biomass really mean for broadband albedo? And what aspects of the biomass matter most for albedo?

Figure 3: idem, but with a panel showing broadband albedo as a function of biomass concentration, for 3 cell radii.

Figure 4: idem, but with a panel showing broadband albedo as a function of layer thickness, for different mass concentrations.

---

## Author Comment (AC2) · 27 Jun 2017

Dear Reviewer,

thank you very much for your comments. I have been through them in detail and made amends as requested. The revised manuscript with the changes will be made available after the other reviewers have commented. For now, I can provide a line-by-line response detailing the changes made to the manuscript and present new versions of the figures I have enhanced according to your suggestions. I have included the full

figure captions at the bottom of this page, beneath my responses.

Responses:

Main Comment:

This crystal clear, relevant, and very complete paper provides a physically-based framework for quantifying the effect of biological activity on the albedo of snow and ice. For a radiative transfer expert, this paper dwells quite long on RT theory and observational techniques. But I guess that the authors have a much broader audience in mind for this paper (but this is nowhere stated explicitly!), including biologists that have not worked much with RT before. In that sense, this manuscript is a great onestop reference for everyone working on bioalbedo. I have only one major request: For figures 2, 3, and 4, could an additional panel be added showing the dependence of the broadband albedo on the variable of interest? Otherwise, I only have some minor suggestions for improvement. Note to the editor: my background is in radiative transfer modelling, so separate advice on the biological aspect of this paper should be sought by another reviewer.

Response to main comment:

Thank you! We have added a note in the abstract to indicate the intended audience. We have updated our figures 2, 3 and 4 to include inset panels with broadband albedo against variable of interest. We thank the reviewer for this suggestion and agree that the result is a more informative plot.

Specific Comments:

P1L28-31 : I would cast some of the 10 challenges differently, such that it is clear what the challenge exactly is. E.g. "Ambiguity in terminology: -> "Reconciling ambiguous terminology"; "Surface anisotropy" -> "Accounting for surface anisotropy"; "Measurement and instrument configurations" -> "Standardizing measurement and instrument configurations" or similar.

[Figure]

Amended throughout

P1L37: ... on THE ice surface ... P4L19: You use both BioSNICAR and Bio-SNICAR (with and without hyphen). Please standardize.

Amended throughout.

P4L21: Please turn this list into a table, with an additional column showing the mathematical symbol used in this paper. Instead of the somewhat cumbersome lines 42-45. P4L21: I guess that these library files are wavelength-dependent? At least, the items 2,3 and 4 in the list? Could you indicate which information is wavelength-dependent?

These two comments have been addressed together by creating a new table (Table 1) and referring to it in the main text.

P5L39: This is a good example of a section that I would recommend to shorten significantly, if the audience of this paper were strictly limited to RT specialists. No new insights are presented here, and a paper like Schaepman-Strub et al. covers this entire section.

I agree, but the audience is broader than RS specialists,. It is specifically intended for glaciologists and biologists who might benefit from an overview of the key literature. This has now been made explicit in the abstract and introduction.

P7L15: strictly speaking, the rightmost part of the equation is a definition of r_eff. You could add a second equation here, defining r_eff as r_eff = 3 / (rho_ice * SSA)

Amended

P10L11: Ambiguous. Please reformulate: "... unless dust can be accounted for accurately. Otherwise, there is a high risk of biomarker false positive"

Amended

P10L12: on the other hand, rough surfaces create photon cavities that increase scattering and absorption, and lower albedo (e.g., Cathles et al., 2011, Ann. Glac. 52(59))

Amended and citation added to text and reference list.

P13L14: straightforward

Amended

Figure 2: It would be very illustrative to add a panel here that shows broadband albedo as a function of pigmentation level. I understand that this requires an additional setting, namely the prescription of an atmospheric vertical profile determining the spectral composition of the radiation arriving at the surface. Nonetheless, I believe that this would be very instructive. How much does the presence of biomass really mean for broadband albedo? And what aspects of the biomass matter most for albedo?

We have added broadband subplots of broadband albedo against biomass for each pigment scenario to Figure 2. We have also added a new figure to Appendix 2 showing broadband albedo against pigmentation. The plot shows that secondary carotenoid production is important for determining biological albedo reduction, but that above a threshold increasing carotenoid mass fraction has diminishing effect on the light absorbing properties of the cells. As biomass concentration increases, changing pigmentation has a greater effect on albedo.

Figure 3: idem, but with a panel showing broadband albedo as a function of biomass concentration, for 3 cell radii.

Added

Figure 4: idem, but with a panel showing broadband albedo as a function of layer thickness, for different mass concentrations

Added

New Figure Captions:

Figure 2: Spectral albedo of snow (grain radius 1500 $\mu$m) with equal biomass loading of algal cells with varying pigmentation. In all four simulations, chlorophyll a = 1.5% total cell dry weight. In A) Primary and Secondary carotenoids = 10% dry weight each. In B) primary and second carotenoids = 5% dry weight each. In C) primary and secondary carotenoids = 1% dry weight each. In D) no carotenoids are present, the cell contains chlorophyll only. In all simulations the solar zenith was 60°. The legend applies to all four subplots. Inset plots show broadband albedo against biomass concentration for each pigment mixture.

Figure 3: Simulations of 1500 $\mu$m radius ice grains with no interstitial water or inorganic impurities and biomass concentrations 0.01, 0.1, 0.5, 1 and 2 mgalg/gsnow confined to a thin (3 mm) surface layer. The mass fraction (% dry weight) of pigments in the cells was 1.5% for chlorophyll a and 5% for each of primary and secondary carotenoids. In A) the cell radius was 5 $\mu$m, in B) the cell radius was 15 $\mu$m and in C) the cell radius was 25 $\mu$m. In all plots the solar zenith was 60°. Legend applies to all three subplots. Insets show broadband albedo against biomass concentration for each cell size.

Figure 4: A constant biomass (0.5 mgalgae/ gice, pigment mass fractions (% total cell dry mass) = 1.5% chlorophyll a, 5% primary and secondary carotenoids, 15$\mu$m cell radius) distributed vertically in layers of ice (1500 $\mu$m grain radius) of varying thickness (1, 2, 3, 4, 5, 10 mm). B) Varying concentrations of mineral dust in a 3 mm surface layer (0.1, 0.5, 1, 1.5, 2 mgdust/ gice) on otherwise clean ice (grain radius 1500 $\mu$m). The dust used was SNICAR's 'dust 4' which has grain radii 2.5 – 5 $\mu$m; C) Equal mass concentrations (0.01, 0.1, 0.5, 1, 2 mgimpurity/ gice) of algal cells (pigment mass fractions (% total cell dry mass) = 1.5% chlorophyll a, 5% primary and secondary carotenoids, 15 $\mu$m cell radius) and mineral dust (SNICAR's 'dust 4' which has grain radii 2.5 - 5 $\mu$m) in a 3 mm surface layer in otherwise clean ice (1500 $\mu$m grain radius); D) Albedo of a dry snowpack (grain radii = 1000 $\mu$m) and snowpacks with liquid water as a coating around the ice grains. The legend indicates the thickness of water layer around a 1000 $\mu$m ice grain. Insets show broadband albedo plotted against the relevant model

variable.

[Figure]

[Figure]

**Fig. 1.** Figure 2: Updated

[Figure]

**Fig. 2.** Figure 3: Updated

[Figure]

[Figure]

[Figure]

**Fig. 3.** Figure 4: Updated

[Figure]

**Fig. 4.** Appendix 2: New Figure

---

## Referee Comment (RC2) · S. M. Skiles (Referee) · 28 Jul 2017

This is well written and clear paper that presents a new optical property model/optical property look up table for biological contituents for the snow radiative transfer model SNICAR. I believe this is the novelty and strength of this paper, and the reason I am suggesting major revision is that I believe the authors should focus on this aspect as opposed to in length 'review' they give of general snow spectroscopy/radiative transfer.

Although it is true that field spectroscopy measurements (and the understanding of

snow radiative transfer) do present uncertainty, the authors themselves do not actually present any spectral (or broadband) measurements made in the field, which is puzzling. It is not clear to me if they tried, and felt they were subject to too much uncertainty to publish, or if they did not try because they assumed the uncertainties in these measurements were too large, or maybe the measurements didn't validate the model well enough- whichever it was, the jump from the theoretical radiative transfer modeling for bioalbedo to the general review to snow spectroscopy/optics/terminology feels like two different papers. They state this section is 'a critique of empirical bioalbedo studies' which is not really the case. I can appreciate the desire to point out to others in the research community these uncertainties, but it seems out of place, is not an exhaustive enough review as to act as an only resource for readers, and all of the information in sections 4.1, 4.2, 4.3, 4.8, and 4.10 can be found in a wide range of publications over multiple decades. I suggest the authors point to the numerous papers on general snow albedo and radiative transfer, and make the paper more concise and novel. I do think this is a good paper, but it could be improved by focusing the uncertainty discussion on those uncertainties that actually pertain to the adaption of SNICAR to model bioalbedo (sections 4.4, 4.5, 4.6, 4.7, 4.9).

I have a few more general comments, and minor line by line comments are below. -The authors should discuss in more detail how this paper varies from Cook et al. 2017, 'A predictive model for the spectral "bioalbedo" of snow'. -Since no field measurements are presented to validate the outputs, it should be made clear that this is a purely theoretical model. Thereby, I would be cautious in generically stating that this can extend ground measurements to remote sensing measurements. Remote sensing of reduction in snow albedo due specifically to particulates is difficult due to lacking spectral/spatial/and-or temporal resolution and has only been carried out successfully in very heavily contaminated regions from spaceborne platforms (I refer you to Warren, 2013(doi:10.1029/2012JD018476))- and even with those retrievals there are subject to high uncertainties. It is different with airborne imaging spectroscopy, but those flights (like the one used by Painter et al., 2001, Painter et al., 2013, and Seidel et al., 2016)

[Figure]

tend to be 'one-off' flights. -Is biological matter on snow and ice actually an impurity? This has been used commonly by the light absorbing community in the past, including myself, but it has begun to move away from it. I would consider using another term, like particulate or constituent.

Line by line:

P1, 16- Has this been recognized? I'm not sure. Also, I would generally avoid using the word significant unless there are statistics to back it up.

P1, 41- There are many more, and perhaps better, citations than the ones used- namely the 2007 and 2013 IPCC reports and Hansen and Nazarenko, 2004

P2, 6- High Sierra –> Sierra Nevada

P2, 7- The retrieval is an optically equivalent grain size.

P2, 37- Can you please provide an estimate of area to back up 'wide spatial coverage'? The Greenland Ice Sheet is massive, and the dark snow/transition zone where biological constituents are found, and impact albedo in a meaningful way, is only along ice sheet margins. (the albedo map here - http://nsidc.org/greenland-today/2013/06/springtime-melt-in-greenland-late-start-rapid-spread/ - comes to mind)

P3, 9- dusts –> mineral dust or simply 'dust'

P3, 12-14- Reword this sentence for clarity. The optical properties of the ice grains mostly matter in the NIR where ice is absorptive, in the visible wavelengths the vertical distribution matters because ice absorption is very weak and scattering is high- which means light can penetrate the snow pack. Still impurities only impact albedo in the near surface (maybe at maximum 30 cm, I have found it to be closer to 6-10 cm for aging snow in the presence of dust) because the imaginary index refraction increases across the visible spectrum.

P3, 18-20- Physical modeling of snow radiative transfer has been applied to remote

sensing retrievals over snow and ice for decades- I'm not sure what you are saying here.

P3, 41- A good citation here would be Kaspari et al., 2015 (10.1002/2014JD022676). But my response to this was- are they well-validated for contaminated snow? Or do they just decently approximate in a few instances? I would revisit these studies to see if they actually validate the model in the presence of black carbon and/or dust (Painter et al., 2007 estimated the partitioning between dust and black carbon, and the black carbon concentrations used were far far above what we have ever actually measured at that site)? The fact that SNICAR could not represent snow albedo in the present of dust prompted the development of a method to retrieve the optical properties of dust in snow, see Skiles et al., 2016 (doi: 10.1017/jog.2016.126).

P4, 8- This review (a generous term in regards to black carbon and dust) is over seven years old in a rapidly advancing field, certainly there are more up to date citations in addition to Gardner and Sharp? Just to point out they use black carbon concentrations from Hansen and Nazarenko, 2004 after which there has been an explosion of studies documenting black carbon concentrations, and the impact of dust and black carbon on snow albedo.

P4,12- Be explicit about what you mean here in terms of the optics

P5,5- Does this matter for snow albedo? Is not included because SNICAR is only over the solar wavelengths? If there is a reason you felt it important to mention this, support the statement.

P5, ~14- I think it is worth noting at the end of the description here that similar methodology has been used to infer the imaginary part of refractive index of sea-ice sediment by Light et al (1998), for hydrated salt crystals in sea ice by Carns et al. (2016), and for dust in snow by Skiles et al. (2016).

Light B, Eicken H, Maykut G and Grenfell T (1998) The effect of included particulates

on the spectral albedo of sea ice. J. Geophys. Res., 103(C12), 27739–27752

Carns RC, Light B and Warren SG (2016) The spectral albedo of sea ice and salt crusts on the tropical ocean of Snowball Earth: II. Optical modeling. J. Geophys. Res.: Oceans, 121(7), 5217– 5230

P5, 21- I thought there were four options for spectral irradiance in SNICAR? Midlatitude clear or cloudy and then Greenland clear or cloudy.

P5, 25- These are different from the default look up table in SNICAR, which has the same range of radii? Can you give more details about how these were produced?

I am refraining from giving comments for section 4.1, the general theme was that you can find this information well documented elsewhere, namely in Schaepman-Strub et al., 2006

P7, 27- If you keep this section, again I think the information can be found in many other places, I think it might be worth mentioning here that TARTES offers a range of options for grain size/shape

P7, 35- I think you should explicitly make it clear that observable grain size is very different than optical grain size- observable grain sizes will lead to high errors because they represent what we see, not the absorbing path length

Section 4.3 I suggest removing

P9,28- Are these observations, or just model results? The plots in Warren and Wiscombe, 1980 are model results- and it makes sense yours would agree with those and with Flanner, 2007 because it is the same basic theory. I would suggest reading 3.2.2 of Skiles et al., 2016 where we found that SNICAR did model the snowpack this way, but observations do not show this effect (full spectal albedo time series can be found in Skiles and Painter, 2016 (doi:10.1017/jog.2016.125)), even under extremely heaving dust loading. If biological constituents are heterogeneously mixed among snow grains, ice absorption may still be dominant even with additional biomass.

P10, 35- Can you justify why it may be important to isolate the bioalbedo here, rather than later in the section? In the context of climate and hydrologic modeling it is the integrated impact of all impurities that matters, so it is helpful to point out why you may want to separate the unique reduction in albedo due to just biological constituents.

p11, 15- It is important to point out here that dust optical properties are highly variable, and SNICAR uses by default a 'global mean' dust optical property representation- you also only specified dust in the largest particle size bin in your runs, which is the least absorbing and probably an unrealistic representation of the dust particle size distribution anywhere... also, what about black carbon? It seems to be missing in this discussion, yet studies have found that in relatively high concentrations at the ice sheet edges.

Section 4.8 I suggest removing, information also found elsewhere. if it stays, it seems very out of place here and should be up with other radiative transfer stuff. Also, it is possible to set up an experiment to measure this if you have a spectrometer. If you are truly interested in this I recommend reading Tom Painter's dissertation (UCSB, 2000).

P11, 38- Clarify and give a citation for 'preferentially melt smaller grains with larger SSA's'

P11, 40- Albedo is also reduced in the NIR

Section 4.10 I would suggest removing, it is out place, especially since you do not present albedo measurements in the paper. And the gist of you paper is to model the spectral shape of snow albedo containing biological constituents, or in turn, inverting spectral reflectance to back out biological constituents, so it is unclear to me in what case broadband albedo would be sufficient in this context. You also seem to represent hemispherical reflectance as a 'decent' approximation of albedo, which is not the case, a model is required to go from a reflectance measurement to an albedo (this is described a number of places- I described it recently in Casey et al., 2017 (doi: 10.1002/2016JD026418) .

Figures-

Please make sure all of the information in figure captions is also in the paper (methods, etc.)

Figure 1: I would keep part B, and refer readers to papers containing other figures

Figure 5: Unnecessary, would be more appropriate if you had presented albedo measurements in this paper.

Metadata sheets would be more appropriately included if you had presented albedo measurements in this paper.

Please do not hesitate to get in touch with questions or comments- McKenzie Skiles, m.skiles@geog.utah.edu

---

## Author Comment (AC3) · 15 Aug 2017

Dear Dr Skiles,

thanks for reviewing our paper, please find our response in the pdf supplement

Best wishes

Joseph Cook

[Figure]

Please also note the supplement to this comment:
https://www.the-cryosphere-discuss.net/tc-2017-73/tc-2017-73-AC3-supplement.pdf

---

## Author Response (AR1)

Dear Editor,

We have now responded to two sets of reviewer comments for our manuscript 'Quantifying bioalbedo: A new physically-based model and discussion of empirical methods for characterizing biological influence on ice and snow albedo'. Both reviewers responded positively to the manuscript overall and we have addressed each comment in turn. We provide here a revised manuscript altered in accordance with the review comments, an additional 'marked up' manuscript, and a point-by-point response to both reviewers (included in this PDF).

We were very happy to amend our manuscript according to the comments and consider the manuscript to have improved greatly as a result of the thorough and insightful work of both reviewers; however, there was one point of contention. Reviewer 2 requested that the 'review' elements of the paper be removed, which is understandable given her high level of expertise in field spectroscopy and radiative transfer. However, based on positive feedback from the bioalbedo community as well as positive comments from reviewer 1 regarding the content and structure of the manuscript, we feel strongly that the wider discussion of empirical methods is a crucial aspect that enhances the reach and impact of the work. We set out to write a paper that not only provided a new radiative transfer scheme for quantifying biological albedo reduction but also offered synthesis and clarity to bioalbedo researchers lacking specific expertise in spectroscopy and/or radiative transfer. We feel that this paper achieves this and will therefore make a more useful addition to the literature than a modelling report.

We hope that the responses provided and the amends made to the revised manuscript are sufficient to proceed with publication of our paper.

Thank you again for handling our manuscript,

Best Wishes

Joseph Cook

15/08/2017

**Response to Reviewer 1:**

This crystal clear, relevant, and very complete paper provides a physically-based framework for quantifying the effect of biological activity on the albedo of snow and ice. For a radiative transfer expert, this paper dwells quite long on RT theory and observational techniques. But I guess that the authors have a much broader audience in mind for this paper (but this is nowhere stated explicitly!), including biologists that have not worked much with RT before. In that sense, this manuscript is a great onestop reference for everyone working on bioalbedo. I have only one major request: For figures 2, 3, and 4, could an additional panel be added showing the dependence of the broadband albedo on the variable of interest? Otherwise, I only have some minor suggestions for improvement. Note to the editor: my background is in radiative transfer modelling, so separate advice on the biological aspect of this paper should be sought by another reviewer.

Thank you! We have added a note in the abstract to indicate the intended audience. We have updated our figures 2, 3 and 4 to include inset panels with broadband albedo against variable of interest. We thank the reviewer for this suggestion and agree that the result is a more informative plot.

P1L28-31 : I would cast some of the 10 challenges differently, such that it is clear what the challenge exactly is. E.g. "Ambiguity in terminology: -> "Reconciling ambiguous terminology"; "Surface anisotropy" -> "Accounting for surface anisotropy"; "Measurement and instrument configurations" -> "Standardizing measurement and instrument configurations" or similar.

Amended throughout

P1L37: ... on THE ice surface ... P4L19: You use both BioSNICAR and Bio-SNICAR (with and without hyphen). Please standardize.

Amended throughout.

P4L21: Please turn this list into a table, with an additional column showing the mathematical symbol used in this paper. Instead of the somewhat cumbersome lines 42-45.

P4L21: I guess that these library files are wavelength-dependent? At least, the items 2,3 and 4 in the list? Could you indicate which information is wavelength-dependent?

These two comments have been addressed together by creating a new table (Table 1) and referring to it in the main text.

P5L39: This is a good example of a section that I would recommend to shorten significantly, if the audience of this paper were strictly limited to RT specialists. No new insights are presented here, and a paper like Schaepman-Strub et al. covers this entire section.

I agree, but the audience is broader than RS specialists,. It is specifically intended for glaciologists and biologists who might benefit from an overview of the key literature. This has now been made explicit in the abstract and introduction.

P7L15: strictly speaking, the rightmost part of the equation is a definition of r_eff. You could add a second equation here, defining r_eff as r_eff = 3 / (rho_ice * SSA)

Amended

P10L11: Ambiguous. Please reformulate: "... unless dust can be accounted for accurately. Otherwise, there is a high risk of biomarker false positive"

Amended

P10L12: on the other hand, rough surfaces create photon cavities that increase scattering and absorption, and lower albedo (e.g., Cathles et al., 2011, Ann. Glac. 52(59))

Amended and citation added to text and reference list.

P13L14: straightforward

Amended

Figure 2: It would be very illustrative to add a panel here that shows broadband albedo as a function of pigmentation level. I understand that this requires an additional setting, namely the prescription of an atmospheric vertical profile determining the spectral composition of the radiation arriving at the surface. Nonetheless, I believe that this would be very instructive. How much does the presence of biomass really mean for broadband albedo? And what aspects of the biomass matter most for albedo?

We have added broadband subplots of broadband albedo against biomass for each pigment scenario to Figure 2. We have also added a new figure to Appendix 2 showing broadband albedo against pigmentation. The plot shows that secondary carotenoid production is important for determining biological albedo reduction, but that above a threshold increasing carotenoid mass fraction has diminishing effect on the light absorbing properties of the cells. As biomass concentration increases, changing pigmentation has a greater effect on albedo.

Figure 3: idem, but with a panel showing broadband albedo as a function of biomass concentration, for 3 cell radii.

Added

Figure 4: idem, but with a panel showing broadband albedo as a function of layer thickness, for different mass concentrations

Added

[Figure]

Amended Figure 2

[Figure]

Amended Figure 3

[Figure]

Amended Figure 4

New Fig A2

**Detailed Response to Reviewer 2:**

P1 L16: I would argue that this is well established. However, I have removed the word 'significant' as I agree it is ambiguous in terms of statistical significance.

P1 L41: Added citation to IPCC 2013 and Hansen and Nazarenko, 2004 as requested.

P2 L6: Changed "High Sierra" to "Sierra Nevada" as requested

P2 L7: Added "optically equivalent" as requested

P2 L37: Clarified on L37 that we are referring to ablating areas of the Greenland Ice Sheet and added the range of total bare ice extent for the past 15 years (citing Shimada et al., 2016) as follows: "The wide spatial coverage of algae on the ablating areas of the Greenland Ice Sheet (which, over the past 15 years have comprised between 5 – 16% of the entire ice sheet area: Shimada et al., 2016)"

P3 L9: Added "mineral" as requested

P3 L12-14: Reworded for clarity. Now reads: "The vertical position of light absorbing impurities in the shallow-subsurface is also important because in the visible wavelengths absorption by ice crystals is low (Warren, 1982), meaning light can penetrate the snowpack."

P3 18-20: Yes, fair enough. I appreciate that this could be misconstrued and is arguably tangential to our paper. Sentence removed.

P3 L41: Thanks for the suggestions, I have amended the sentence to read: "These models are well-validated for clean snow (Grenfell et al., 1994) and have been applied to snow contaminated with black carbon and dust (Painter et al., 2007, Flanner et al., 2007; Gardner and Sharp, 2010, Brandt et al., 2011; Kaspari et al., 2015), although their performance for impurity-laden snow has been questioned (Skiles et al., 2016)."

P4 L8: New citations added for black carbon in snow: Deems et al., 2013; Kaspari et al., 2014; Skiles et al., 2017)

P4 L12: Amended for clarity: "Both models characterized biological impurities simplistically by updating the absorption terms in existing inorganic impurity models rather than predicting the full optical properties of cells using Lorenz-Mie calculations."

P5 L5: Deleted. I originally mentioned this because UV absorbing pigments are known to be produced by ice algae, so there is some biological relevance.

P5 L4: Added the following sentence to the m/s: "Similar methodologies have previously been employed to infer the imaginary part of the refractive index of sediments (Light et al, 1998) and hydrated salt crystals (Carns et al., 2016) in sea ice and for dust in snow (Skiles et al., 2017)."

P5 L21: Yes, I agree this reads ambiguously. Now updated for clarity.

P5 25: The SNICAR code I used did not have so wide a range of ice grain sizes as this, so the larger grains were produced using a Mie code and Warren and Wiscombe's data for ice optical properties.

P7 L27: Added the following text and citations to P8 L14: "Non-spherical grain shapes can influence the surface albedo (Dang et al., 2016; Libois et al., 2013) and can be simulated in the radiative transfer scheme TARTES (Libois et al., 2013)."

P7 L35: Thanks, I have added the following text to P8 L14: "It is also important to note that optical grain size and observable grain size are different properties and high uncertainty results from assuming that they are equivalent – this is because the observable grain size does not represent the absorbing path length."

P9 L28: These are model results not observations. We do not yet have sufficient cell count or pigmentation data to link biological growth with albedo decline (although this data is forthcoming and will comprise a separate paper) except to say that areas of higher biomass were associated with lower albedo and characteristic spectral shapes. We have acknowledged the uncertainty described in your papers on dust in snow: "Skiles et al., 2016 and Skiles and Painter (2016) found that modelled albedo decline due to dust and black carbon in snow was dissimilar to measurements, indicating the urgent need for radiative

transfer modelling of biological impurities to be field validated. We have recently undertaken field work for this purpose and will publish the outcomes in a future paper."

P10 L33: added "Isolating the biological contribution to snow and ice albedo is crucial for modelling ice dynamics under future climate scenarios where the intensity and spatial coverage of algal blooms will likely vary differently to inorganic impurities."

P11 L15: Text added: "This was found to be true for three other classes of dust (size fractions 0.05-0.5 um, 0.5-1.25um, 1.25-2.5 um) although the data are not shown here. Real dust optical properties vary widely and SNICAR uses a global mean that is unlikely to be a good representation of real dusts in any specific location."

We have opted not to discuss BC in detail in this paper for two reasons: 1) it is documented thoroughly in many other recent papers and our focus is really on biological impurities; 2) one of the central questions in the bioalbedo community is "what darkens the Greenland Ice Sheet (and other ice masses)" - the two main hypotheses are algal blooms and melt-out of Holocene dusts. Therefore we have focused our discussion on these materials. Still, we agree that BC needs to be mentioned here, so we have added the following: "In our simulations black carbon was found to be a more effective albedo reducer than algal cells per unit mass (data not shown)."

Section 4.8 We would like to keep this section as we suspect it is one of the less well understood aspects of spectroscopy. While this is elementary to you and your peer group, this is not obvious information for those tackling bioalbedo from a biological or glaciological background and therefore bears repeating, with direction to appropriate supporting literature, in this paper.

P11, 38- I have deleted the text 'preferentially melt smaller grains with larger SSA's'

P11 L40: "visible spectrum" amended to "solar spectrum"

Section 4.10: Again, we do not want to remove this section because we feel it is important for future bioalbedo work. We have added further clarifying information about the relationship between hemispheric reflectance and albedo and cited your work in Casey et al (2017) as requested. I'm surprised that the inclusion of broadband albedo measurements is unclear to you given its importance for determining energy balance. Many studies will measure albedo of impurity laden ice only to estimate the effect of impurity loading on energy balance and will only want broadband measurements. I also expect that many of the end users of BioSNICAR will discard the spectral information in favour of broadband albedo. We don't really think we have equated reflectance with albedo, having explicitly stated that to estimate albedo directional/conical reflectance must be integrated over the viewing hemisphere taking into account the ARF in three places in the manuscript (pg 7 and section 4.8 and section 4.10); however, we appreciate that we could have been clearer so we have updated the discussion to include the following text and citation of your paper:

"... to determine the reflectance properties of a surface component with dimensions of the order of decimetres (e.g. an algal bloom), a hemispheric-conical reflectance measurements (HCR: Fig. 1A) made using a sensor with a limited field of view are likely more appropriate and can be used to estimate the albedo by integrating over the entire viewing hemisphere, accounting for anisotropic scattering."

"With knowledge of ARFs for the sample surface and reflectance panel, these measurements can be used in conjunction with an angular integration model to approximate albedo (Grenfell, 2011; Casey et al., 2016)"

We have also provided a new figure (Figure 6) which demonstrates the difference between albedo and HCRF measurements for the same two sites, and discussed it in the following text:

"The variation between albedo measured using a cosine collector and reflectance measured using an 8° collimating lens at nadir view for the same surfaces (clean ice and algal ice near the IMAU Weather station 'S6' on the southwest Greenland Ice Sheet) is demonstrated in Figure 6. The measurements were all made within 15 minutes during constant clear sky conditions on 14th July 2017 using the same spectral radiometer, fibre optic and tripod arrangement that did not move between measurements. The variation is most likely due to spatial integration of a wider range of surfaces for the cosine collector and the lack of angular integration for the collimating lens."

**Additional amends:**

The term 'critique' has been changed to 'discussion' throughout the manuscript to better represent the aims and scope of the paper in response to reviewer comment. This includes the paper's title.

We added an explicit mention of the target audience to make clear the intended use of the paper and justify the review sections: "
[revised manuscript text omitted]

**Figure 1: The logo of Copernicus Publications.**

---

## Author Response (AR2)

Joseph Cook

University of Sheffield

Department of Geography

Sheffield

UK, S10 2TN

Joe.cook@sheffield.ac.uk

**Authors Response:**

Dear Editors,

We are delighted to hear that our paper will be published in The Cryosphere and wish to thank the editorial team and the reviewers for their support in revising the manuscript. Also, we are sincerely grateful to the reviewers who offered insightful and thorough comments and greatly improved the original manuscript. We are very grateful for their service.

We confirm that the requested technical corrections have been addressed in the most recent file upload.

Kind regards,

Joseph Cook and co-authors